# The inositol 5-phosphatase INPP5B regulates B cell receptor clustering and signaling

Alaa Droubi[1], Connor Wallis[1], Karen E. Anderson[2], Saifur Rahman[3], Aloka de Sa[1], Taufiq Rahman[3], Len R. Stephens[2], Philip T. Hawkins[2], and Martin Lowe[1]

Upon antigen binding, the B cell receptor (BCR) undergoes clustering to form a signalosome that propagates downstream signaling required for normal B cell development and physiology. BCR clustering is dependent on remodeling of the cortical actin network, but the mechanisms that regulate actin remodeling in this context remain poorly defined. In this study, we identify the inositol 5-phosphatase INPP5B as a key regulator of actin remodeling, BCR clustering, and downstream signaling in antigen-stimulated B cells. INPP5B acts via dephosphorylation of the inositol lipid $PI(4,5)P_2$ that in turn is necessary for actin disassembly, BCR mobilization, and cell spreading on immobilized surface antigen. These effects can be explained by increased actin severing by cofilin and loss of actin linking to the plasma membrane by ezrin, both of which are sensitive to INPP5B-dependent $PI(4,5)P_2$ hydrolysis. INPP5B is therefore a new player in BCR signaling and may represent an attractive target for treatment of B cell malignancies caused by aberrant BCR signaling.

## Introduction

B cells are a key component of the adaptive immune system, responsible for the production of antibodies, antigen presentation, and cytokine secretion that together help mediate and sustain an immune response (LeBien and Tedder, 2008). B cell development and function are critically dependent on the appropriate activation of their B cell receptors (BCRs; Yam-Puc et al., 2018). BCR engagement by antigen triggers a sequence of cellular events including receptor clustering, intracellular signaling, and subsequent BCR–antigen receptor complex internalization (Avalos and Ploegh, 2014). The actin cytoskeleton is a key mediator of these events and undergoes a characteristic and dramatic reorganization upon receptor activation (Li et al., 2018; Tolar, 2017). In resting B cells, surface BCRs exist in nanosized clusters that exhibit low lateral mobility due to physical constraint provided by the cortical actin network (Mattila et al., 2013; Treanor et al., 2010), thereby ensuring low levels of tonic signaling. Disassembly of the cortical actin, promoted by an early BCR signal upon receptor engagement (Liu et al., 2011; Liu et al., 2012; Sharma et al., 2009), increases BCR mobility and leads to receptor assembly into microscale clusters. Receptor clustering allows Src-family tyrosine kinases to phosphorylate their immunoreceptor tyrosine-based activation (ITAM) motifs (Rolli et al., 2002; Sohn et al., 2008), and the recruitment of intracellular signaling molecules that leads to the activation of multiple signaling pathways including the Ras-MAPK, phosphoinositide 3-kinase (PI3K), and phospholipase

Cγ2 (PLCγ2) pathways (Dal Porto et al., 2004). BCR microclusters further grow to form a central platform for signal amplification as well as immunocomplex endocytosis (Fleire et al., 2006). The growth of BCR microclusters is mediated by a PLCγ2-regulated phosphatidylinositol 4,5-bisphosphate $(PI(4,5)P_2)$ gradient across their boundaries (Xu et al., 2017) as well as the phosphatidylinositol (3,4,5)-trisphosphate $(PI(3,4,5)P_3)$-dependent activation of the Rac–guanine nucleotide exchange factor (GEF) DOCK2 (Wang et al., 2017). Therefore, tight control of these inositol lipids is critical for such a highly dynamic cellular event.

BCR-mediated actin reorganization requires transient deactivation of ezrin, reducing the linking of cortical actin to the plasma membrane (Gupta et al., 2006; Treanor et al., 2011), accompanied by cofilin-mediated actin severing (Freeman et al., 2011). Following initial actin disassembly to allow receptor mobilization, actin polymerization propagates in the vicinity of BCR microclusters (Liu et al., 2012) to stabilize the central cluster. The roles of different actin remodeling machineries, including Vav (Treanor et al., 2011; Weber et al., 2008), CDC42 (Burbage et al., 2015; Gerasimcik et al., 2015; Guo et al., 2009), Wiskott-Aldrich syndrome protein (WASP; Huang et al., 2020; Rey-Suarez et al., 2020), and Arp2/3 (Bolger-Munro et al., 2019) in BCR clustering and activation have been established. However, little is known about the molecular events that trigger the initial actin remodeling that occurs upon receptor activation. Ezrin and cofilin are key players, and like many other actin regulators,

........................................................................................................................

[1]Faculty of Biology, Medicine and Health, University of Manchester, Manchester, UK; [2]Inositide Laboratory, Babraham Institute, Cambridge, UK; [3]Department of Pharmacology, University of Cambridge, Cambridge, UK.

Correspondence to Martin Lowe: martin.p.lowe@manchester.ac.uk.

their activity is strongly affected by PI(4,5)P$_2$ (Bosk et al., 2011; van Rheenen et al., 2007), but the degree to which this inositol lipid participates in early BCR mobilization, and the enzymes responsible, are poorly defined. BCR activation results in PI(4,5)P$_2$ hydrolysis via the action of PLCγ$_2$, which converts the lipid into the second messengers inositol triphosphate (IP$_3$) and diacylglycerol (DAG), and this event is well established as a key step in BCR signaling (Wang et al., 2000). It has been proposed that PLCγ$_2$ also plays a role in deactivating ezrin upon BCR activation (Hao et al., 2009; Treanor et al., 2011), but B cells lacking PLCγ$_2$ are still able to initiate BCR clustering, albeit with restricted capacity to grow (Treanor et al., 2011; Wang et al., 2017; Xu et al., 2017). This suggests the existence of other mechanisms that control PI(4,5)P$_2$-dependent actin remodeling during the initial stage of BCR activation and signaling.

The inositol 5-phosphatases oculocerebrorenal syndrome of Lowe (OCRL) and type II inositol polyphosphate 5-phosphatase B (INPP5B) preferentially hydrolyze PI(4,5)P$_2$ to produce phosphatidylinositol 4-phosphate (PI4P; Jefferson and Majerus, 1995; Zhang et al., 1995), although they are also capable of hydrolyzing inositol phosphates Ins(1,4,5)P$_3$ and Ins(1,3,4,5)P$_4$ as well as the inositol lipid PI(3,4,5)P$_3$ (Schmid et al., 2004), at least in vitro. OCRL and INPP5B are both widely expressed and share significant sequence similarity and domain organization. In addition to a central 5-phosphatase domain (Jefferson and Majerus, 1996), they both have an N-terminal PH (Pleckstrin Homology) domain (Mao et al., 2009) and a C-terminal ASH (ASPM-SPD2-Hydin) domain (Ponting, 2006), followed by an inactive Rho-GAP domain (Erdmann et al., 2007). The ASH domain binds Rab GTPases (Hou et al., 2011; Hyvola et al., 2006) and the endosomal adaptor proteins APPL1 (Erdmann et al., 2007) and IPIP27A/B (also known as Ses1/2 or PHETA1/2; Noakes et al., 2011; Swan et al., 2010), whereas the Rho-GAP domain binds the actin regulators Cdc42 and Rac1 (Erdmann et al., 2007; Faucherre et al., 2003). OCRL has binding sites for clathrin machinery (Choudhury et al., 2005; Choudhury et al., 2009; Mao et al., 2009; Nandez et al., 2014; Ungewickell et al., 2004) that are absent in INPP5B (Williams et al., 2007). OCRL, whose mutations cause the rare genetic disorder Lowe syndrome (Attree et al., 1992), has been implicated in a number of cellular functions, including endocytic trafficking, cytokinesis, ciliogenesis, and lysosomal homeostasis (De Matteis et al., 2017; Mehta et al., 2014). In contrast, the cellular functions of INPP5B remain largely unknown. Here, we identify INPP5B as a new player in BCR signaling. INPP5B controls a pool of PI(4,5)P$_2$ responsible for regulating cortical actin dynamics upon receptor stimulation, which is necessary for clustering of the BCR and its downstream signaling. INPP5B may therefore represent a potential target for therapeutic intervention in malignancies caused by aberrant BCR signaling.

## Results

### BCR clustering is impaired and BCR endocytosis is accelerated in INPP5B-depleted cells

To explore the role of INPP5B in BCR activation, we used the genetically tractable chicken B cell line DT40 to generate an inducible degradation system for the endogenous protein. To achieve this, we edited a DT40 cell line that had already been stably transfected with the F-box protein osTIR1 fused to a myc-His tag, to express INPP5B fused to an auxin degron tag (AtIAA17; Nishimura et al., 2009). The absence of a prenylation sequence in chicken INPP5B allowed us to place the tag at the C-terminus preceded by a linker containing a FLAG-His tag for immunodetection (Fig. S1 A). We initially generated an INPP5B$^{Degron/Degron}$ cell line, where both alleles of INPP5B were tagged with the auxin degron, which was confirmed by RT-PCR on cDNA generated from INPP5B$^{Degron/Degron}$ cells (Fig. S1 B). Immunoprecipitation (IP) and Western blotting of protein extracts from INPP5B$^{Degron/Degron}$ cells with anti-FLAG antibody revealed one band, which was absent in INPP5B$^{WT/WT}$ cells (Fig. S1 C). Addition of auxin to the INPP5B$^{Degron/Degron}$ cells resulted in rapid degradation of the tagged protein, which was reduced to ∼5% of the control level within 1 h (Figs. 1 A and S1 C).

The BCR can bind antigens in all possible forms, and multivalent antigens, whether free in solution or presented on a surface, can induce the formation of BCR microclusters (Harwood and Batista, 2010). The formation of BCR clusters occurs at one pole of the cells when cells are stimulated by soluble antigen (Puffer et al., 2007), whereas it forms at the area in contact with antigen-associated surface in the case of surface-presented antigen (Batista et al., 2001). To explore the role of INPP5B in BCR activation, we first examined the BCR response to soluble antigen. INPP5B$^{Degron/Degron}$ cells were exposed to Texas Red–conjugated anti-IgM antibody as a surrogate antigen on ice, warmed, and imaged live by confocal microscopy. Control cells incubated without auxin showed BCR accumulation in caps that were evident in the majority of cells at ∼3 min after stimulation (Fig. 1 B and Video 1). In contrast, ∼70% of the auxin-treated INPP5B-depleted cells failed to form BCR caps up to 5 min after stimulation, and instead the BCR underwent rapid endocytosis shortly after stimulation (Fig. 1 B and Video 1). Using flow cytometry to quantify cell surface levels of BCR, we could confirm the acceleration in the rate of BCR endocytosis in the INPP5B-depleted cells (Fig. 1, C and D). Flow cytometry also confirmed that cell surface BCR levels were comparable in the control and INPP5B-depleted cells before stimulation (Fig. 1 D). Upon endocytosis, the BCR-antigen complex is delivered to lysosomes, where peptides generated by proteolysis of the antigen are loaded onto MHC II molecules for antigen presentation (Hernandez-Perez et al., 2019). Live imaging of cells labeled with LysoTracker confirmed that the internalized BCRs were delivered successfully to lysosomes in both vehicle-treated and auxin-treated cells, consistent with normal post-endocytic traffic of the BCR after INPP5B depletion (Fig. S1 D).

We next examined the ability of INPP5B$^{Degron/Degron}$ cells to form BCR microclusters when exposed to surface antigen using a similar approach to Wang et al. (2017), where cells were stimulated on coverslips coated with anti-IgM antibody as surrogate antigen, and BCR was visualized with monovalent anti-IgM Fab fragments conjugated to a fluorescent dye. Cells were examined by live TIRF microscopy beginning with their initial contact with the antibody-coated coverslips. As expected, during the spreading response of control cells, BCR clusters formed shortly

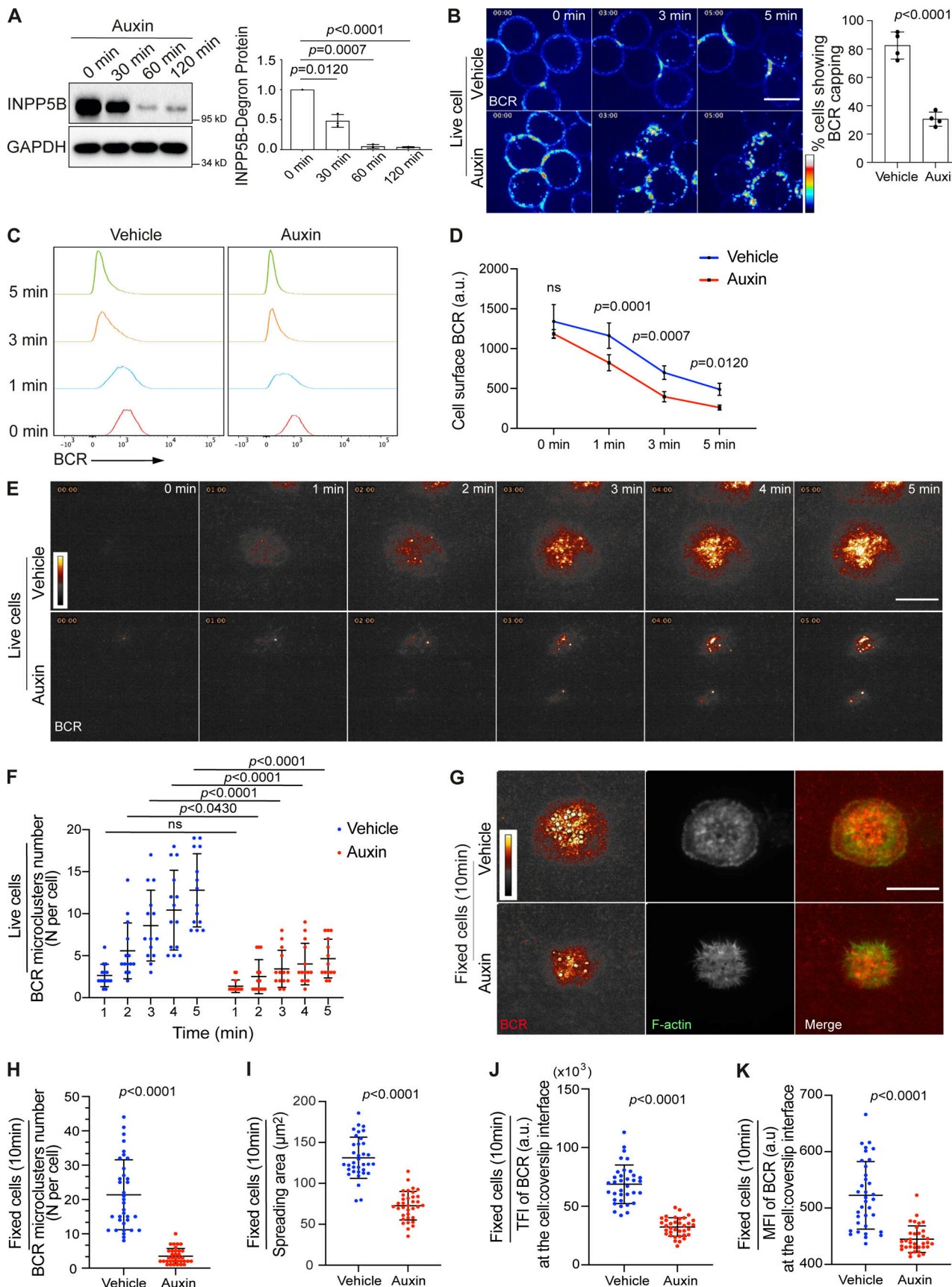

Figure 1. **Impaired BCR clustering and accelerated BCR endocytosis in INPP5B-depleted cells. (A)** INPP5B$^{Degron/Degron}$ cells were treated with auxin as indicated to induce INPP5B degradation, and the remaining INPP5B was immunoprecipitated against its FLAG tag and detected by Western blot using anti poly-

His. Quantification of three such blots by densitometry is shown on the right. Data were analyzed by one-way ANOVA, and P values were calculated using Dunnett's multiple comparisons test. Error bars represent SD. **(B)** Representative confocal images at the indicated time points from INPP5B$^{Degron/Degron}$ cells stimulated with Texas Red–conjugated anti-IgM antibody. Scale bar, 10 μm. See Video 1 for the complete time-lapse confocal images. Quantification of cells showing BCR capping at ~3 min is shown on the right, where a minimum of 50 cells per experiment were examined, and data are expressed as a percentage of cells showing BCR capping events. Data are from four independent experiments. Error bars represent SD, and P value was calculated using Welch's $t$ test. **(C)** Representative histograms at the indicated time points showing the clearance of BCR from the cell surface in INPP5B$^{Degron/Degron}$ cells in response to anti-IgM stimulation. **(D)** Quantification of BCR endocytosis by flow cytometry in INPP5B$^{Degron/Degron}$ cells. Data are from five independent experiments. Error bars represent SD, and P values were calculated using Sidak multiple comparisons test. **(E)** Representative TIRF microscopy images at the indicated time points of BCR clustering in INPP5B$^{Degron/Degron}$ cells that were stimulated on antibody-coated glass. Cells were prelabeled with Fab fragment anti-IgM. See Video 2 for the complete time-lapse TIRF microscopy images. **(F)** Quantification of the number of BCR microclusters per cell is shown in F. Data ($n$ = 15 cells) were analyzed by two-way ANOVA, and P values were calculated using Sidak multiple comparisons test. **(G)** Representative TIRF microscopy images from BCR-labeled NPP5B$^{Degron/Degron}$ cells that were settled on coverslips presenting surrogate antigens for 10 min. F-actin was stained with Phalloidin, and BCR microclusters are denoted by black circles. Scale bar, 10 μm. **(H–K)** Statistical analyses of the number of BCR microclusters per cell, cell spreading area, TFI, and MFI of BCR staining from vehicle-treated and auxin-treated INPP5B$^{Degron/Degron}$ cells in H–K, respectively. The data represent means ± SD of 35 cells. P values were calculated using Welch's $t$ test. Source data are available for this figure: SourceData F1.

after initial contact and continued to increase in number as the cell contact interface increased (Fig. 1, E and F; and Video 2). However, the ability of INPP5B-depleted cells to form BCR microclusters was significantly impaired, as the number of BCR clusters formed during the spreading response was significantly lower (Fig. 1, E and F; and Video 2). To further assess the defect in BCR clustering, BCR-labeled cells were fixed at 10 min and stained with fluorescently conjugated phalloidin to label F-actin to visualize cell spreading. This confirmed that INPP5B-depleted cells had a marked reduction in the total number of BCR microclusters per cell (Fig. 1, G and H), and that this was accompanied by a significant reduction in cell spreading (Fig. 1, G and I). The total fluorescence intensity (TFI) of BCR labeling, as well as the fluorescence intensity divided by contact surface area (mean fluorescence intensity [MFI]), indicative of BCR aggregation at the contact interface, were also significantly lower in INPP5B-depleted cells compared with controls (Fig. 1, J and K). These data indicate that INPP5B is required for normal BCR clustering, and that it is also required for cell spreading on coverslip-bound anti-IgM.

OCRL is closely related to INPP5B and has been proposed to function redundantly with INPP5B during mouse development (Janne et al., 1998) and in various cellular processes (Luo et al., 2013). We therefore analyzed whether depletion of OCRL would impact BCR dynamics. OCRL$^{Degron/Degron}$ cells were generated, using the same strategy as for the INPP5B degron, to allow auxin-mediated depletion of OCRL (Fig. S2 A), and the effect of OCRL depletion on BCR clustering in cells stimulated either in solution or on coverslips was assessed. In both cases, cells lacking OCRL showed normal BCR clustering (Fig. S2, B–E). Cell spreading on coverslip-bound anti-IgM was also unaffected by OCRL depletion (Fig. S2, C and F). These findings indicate that the function of INPP5B in BCR clustering and cell spreading is not shared with OCRL, indicating a distinct requirement for INPP5B in B cells.

#### BCR-mediated signaling is attenuated in INPP5B-depleted cells
Clustering of the BCR is important for the initiation and propagation of a number of downstream signaling pathways (Harwood and Batista, 2010). Therefore, we assessed the impact of INPP5B depletion upon BCR signaling. Three major signaling pathways are activated downstream of the BCR: the

Ras pathway, which leads to extracellular signal–regulated kinase (ERK) activation; the PI3K pathway, which activates Akt; and the PLCγ2 pathway, which mobilizes calcium. In INPP5B-depleted cells, BCR-induced phosphorylation of ERK was greatly attenuated (Fig. 2 A), which was accompanied by reduced BCR-induced Ras activation compared with control (Fig. 2 B). We also observed a delay in the phosphorylation of Akt at Thr308 (human numbering), which is a measure of PI3K signaling, in INPP5B-depleted cells (Fig. 2 A). In contrast, phosphorylation of Akt at Ser473 (human numbering), which is a downstream target of mTORC2, was unchanged (Fig. 2 A). Similar effects upon ERK and Akt phosphorylation were seen with cells stimulated by coverslip-bound anti-IgM (Fig. 2 C).

Phosphorylation of Akt at Thr308 is mediated by PDK1 and depends on PI3K-generated PI(3,4,5)P$_3$ (Currie et al., 1999). PI(3,4,5)P$_3$ levels were therefore quantified by mass spectrometry (MS; Clark et al., 2011), which revealed a significant decrease in total PI(3,4,5)P$_3$ abundance in INPP5B-depleted cells following BCR stimulation, compared with control (Fig. 2 D), consistent with reduced PI3K signaling upon INPP5B depletion. We also analyzed the phosphorylation of CD19, which is critical for the recruitment and activation of PI3K downstream of the BCR (Buhl et al., 1997; Otero et al., 2001). CD19 phosphorylation was significantly compromised in INPP5B-depleted cells (Fig. 2 E), consistent with the observed reduction in P3K activity.

Next, we examined the activation of PLCγ2 pathway in INPP5B-depleted cells, which produces the second messengers DAG and IP$_3$, which in turn mobilizes calcium. DAG levels were measured by MS as a direct measure of PLCγ2 activity. This revealed a significant reduction of DAG mass in INPP5B-depleted cells upon BCR stimulation (Fig. 2 F). We also measured the ability of cells to mobilize calcium in response to BCR stimulation in solution and found that, while the total amount of calcium released was unaltered in INPP5B-depleted cells, the percentage of oscillatory cells and the frequency of calcium oscillations were significantly reduced (Fig. 2, G and H). These data reveal that PLCγ2 activation in response to BCR stimulation is impaired in INPP5B-depleted cells.

Finally, to determine whether depletion of INPP5B impacts proximal signaling, we assessed the phosphorylation of the nonreceptor tyrosine kinase Syk, which is recruited to the ITAMs of CD79A/CD79B upon BCR activation (Wen et al., 2019).

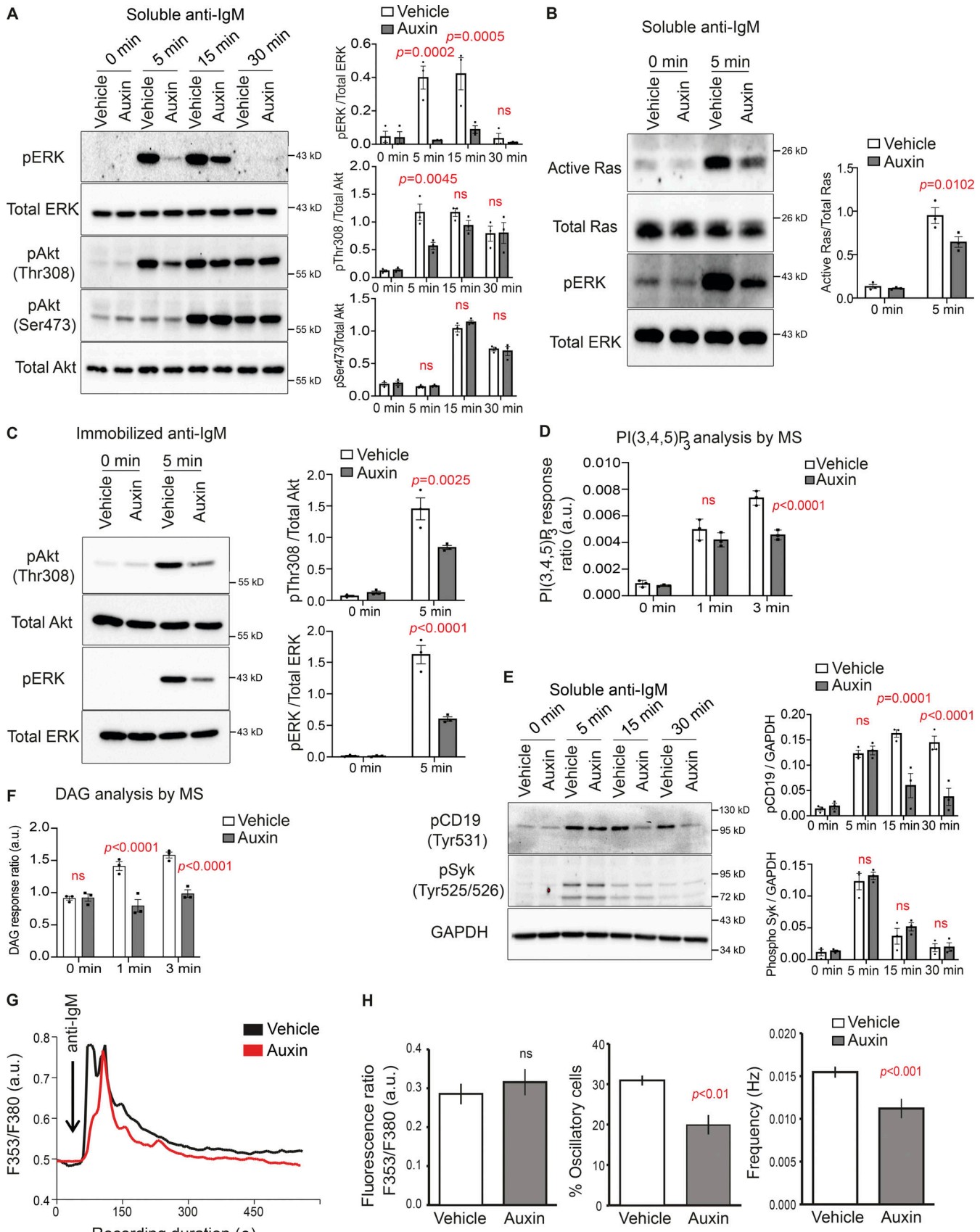

Figure 2. **BCR signaling is attenuated in INPP5B-depleted cells. (A)** Protein extracts from BCR-stimulated INPP5B[Degron/Degron] cells were blotted for ERK and Akt phosphorylation at the indicated time points. Quantification of three such blots by densitometry is shown on the right. Error bars represent SEM. Data

were analyzed by two-way ANOVA, and P values were calculated using Sidak multiple comparisons test. **(B)** GTP-bound Ras was detected using GST-Raf RBD pulldown followed by blotting against Ras. Quantification of three such experiments by densitometry is shown on the right. GTP levels of Ras are expressed relative to total Ras. Error bars represent SEM, and data were analyzed as in A. **(C)** Representative blot from INPP5B$^{Degron/Degron}$ cells that were stimulated on coverslips presenting surrogate antigens for 5 min. Quantification of three such blots is shown on the right. Data were analyzed as in A. **(D)** PI(3,4,5)P$_3$ levels in INPP5B-depleted cells. INPP5B$^{Degron/Degron}$ cells were treated as in A; however, they were stimulated for the times indicated before analysis by MS. PI(3,4,5)P$_3$ levels are expressed relative to PI (see Materials and methods), and data were combined for C36:2, C38:3, and C38:4 species. Error bars represent SD, and data were analyzed as in A. **(E)** Representative pCD19 and pSyk blots from BCR-stimulated INPP5B$^{Degron/Degron}$ cells at the indicated time points. Quantification of three such blots is shown on the right. Data were analyzed as in A. **(F)** DAG levels are expressed relative to internal standard (see Materials and methods), and data were combined for C38:4 and C34:1 species. Error bars represent SEM, and data were analyzed as in A. **(G)** Representative single-cell intracellular Ca$^{+2}$ responses to anti-IgM (added at arrow) in vehicle- and auxin- treated INPP5B$^{Degron/Degron}$ cells. **(H)** Left: Each bar represents peak fluorescence (Fura-2) ratio of anti-IgM response expressed as mean ± SEM ($n \geq 20$ cells from three independent experiments on three different days). Middle: Percentage of cells that displayed spontaneous (Ca$^{2+}$)$_i$ oscillations in total anti-IgM stimulating responding cells. Right: Spontaneous (Ca$^{2+}$)$_i$ oscillation frequency observed in vehicle- and auxin-treated INPP5B$^{Degron/Degron}$ cells. Source data are available for this figure: SourceData F2.

Interestingly, Syk phosphorylation was unaffected by INPP5B depletion (Fig. 2 E). This suggests no effect upon proximal signaling, although the existence of feedback loops may complicate the interpretation of this result (Song et al., 2013). Indeed, CD19 phosphorylation, which is also proximal, was affected by INPP5B depletion, as mentioned above (Fig. 2 E). Together, the data indicate that INPP5B is required for normal BCR-mediated signaling. In contrast, depletion of OCRL had no effect on BCR signaling, as assessed by the extent of ERK and Akt phosphorylation, induced by either soluble or coverslip-bound anti-IgM, consistent with its lack of effect upon BCR clustering (Fig. S2, G–J).

### INPP5B requires its catalytic activity to function in BCR clustering and signaling

To determine whether the function of INPP5B in BCR clustering and signaling is dependent on its catalytic activity, we generated a cell line containing one WT degron-tagged INPP5B allele and one non–degron-tagged allele harboring a G502D mutation that renders it catalytically inactive (Jefferson and Majerus, 1996). As a control, we used the parental cell line containing one degron-tagged and one untagged WT INPP5B allele. This strategy allows degron-tagged INPP5B to be acutely removed from the cells after addition of auxin, leaving cells with either a catalytically inactive enzyme in the case of INPP5B$^{Degron/G502D}$ cells or a WT enzyme in the case of INPP5B$^{Degron/WT}$ control cells. Analysis of genomic DNA extracted from INPP5B$^{Degron/G502D}$ cells confirmed that only the degron-untagged allele was mutated, and RT-PCR confirmed that both alleles of INPP5B in INPP5B$^{Degron/WT}$ and INPP5B$^{Degron/G502D}$ are expressed (Fig. 3 A). BCR expression analysis by flow cytometry confirmed that the receptor is expressed comparably at the surface of both cell lines (Fig. 3 B).

BCR capping was normal in the INPP5B$^{Degron/WT}$ cells upon treatment with auxin, confirming that the single WT allele in these cells is sufficient to maintain INPP5B function (Fig. 3 C, compare with Fig. 1 B). In contrast, in auxin-treated INPP5B$^{Degron/G502D}$ cells, we observed a capping defect similar to that seen upon depletion of INPP5B, which was accompanied by accelerated BCR endocytosis (Fig. 3, C and D). Similarly, upon stimulation with coverslip-bound anti-IgM, both cell spreading (Fig. 3, E and G) and BCR clustering were impaired in cells expressing only catalytically inactive INPP5B (Fig. 3, E, F, and H). The BCR signaling in response to soluble or coverslip-bound anti-IgM was also greatly reduced in auxin-treated

INPP5B$^{Degron/G502D}$ cells, as indicated by reduced phosphorylation of Akt and ERK compared with cells expressing WT INPP5B (Fig. 3, I and J). These data indicate that the inositol phosphatase activity of INPP5B is required for the protein to function in BCR clustering and signaling.

### The dynamics of PI(4,5)P$_2$ metabolism following BCR stimulation is altered in INPP5B-depleted cells

In light of the findings above, and given that INPP5B is mainly a PI(4,5)P$_2$ 5-phosphatase (Jefferson and Majerus, 1995; Schmid et al., 2004), we reasoned that loss of the enzyme would affect PI(4,5)P$_2$ abundance and/or dynamics following BCR stimulation. Using MS to measure the total mass of cellular PIP2 (which will be mostly PI(4,5)P$_2$; Clark et al., 2011) indicated a slight increase in INPP5B-depleted cells stimulated with anti-IgM in solution, although the difference did not reach statistical significance (Fig. S3 A). This is perhaps not unexpected considering that PI(4,5)P$_2$ at the plasma membrane is likely distributed in multiple pools, and any action of INPP5B is likely to be rather localized and affect only a smaller pool of this lipid. We therefore used another way to analyze PI(4,5)P$_2$, namely the established PI(4,5)P$_2$ biosensor EGFP-Tubby PH domain (Szentpetery et al., 2009). The INPP5B$^{Degron/Degron}$ cells were engineered to stably express EGFP-Tubby, and control experiments confirmed that expression of the biosensor did not perturb cell spreading, BCR clustering, or signaling (Fig. S3, B and C).

INPP5B$^{Degron/Degron}$ cells expressing EGFP-Tubby were activated on antibody-coated coverslips and imaged by live-cell TIRF microscopy. During the spreading of control cells, PI(4,5)P$_2$ was present at the initial contact area and appeared enriched in foci upon interacting with the stimulatory surface (Fig. S3 D). PI(4,5)P$_2$ continued to increase in intensity, reaching a maximum ~2 min after contact (Fig. S3 D). As the cell contact area increased, the PI(4,5)P$_2$ foci dissipated, and the lipid appeared to redistribute toward the periphery of spreading cells (Fig. S3 D and Video 3). In INPP5B-depleted cells, PI(4,5)P$_2$ was also enriched in foci at the initial contact area (Fig. S3 D and Video 4). The depleted cells showed reduced spreading (Fig. S3 D and Video 4), as expected, and the abundance of PI(4,5)P$_2$ across the entire contact interface was unchanged (Fig. S3 D), consistent with the lack of effect on total PI(4,5)P$_2$ seen in MS. However, the PI(4,5)P$_2$ foci, formed upon initial contact, persisted for longer in INPP5B-depleted cells (Fig. 4 A), and there was no redistribution of the lipid at the plasma membrane (Figs. 4 A and

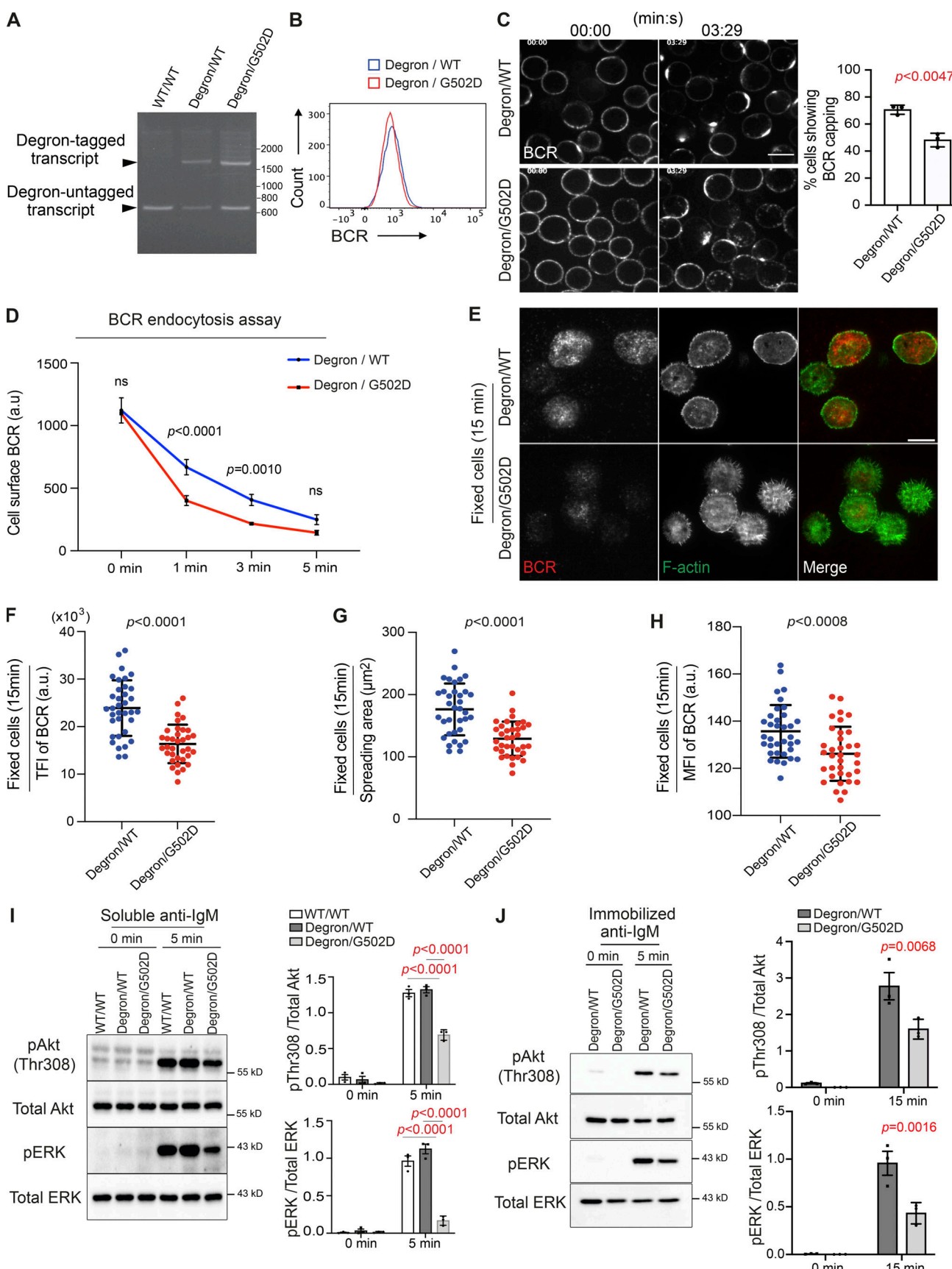

Figure 3. **The impact of INPP5B depletion on BCR clustering and signaling is dependent on the enzyme's catalytic activity. (A)** RT-PCR analysis of the WT and mutant clones using gene-specific primers flanking the *inpp5b* stop codon. **(B)** Analysis of BCR expression in mutant clones by flow cytometry.

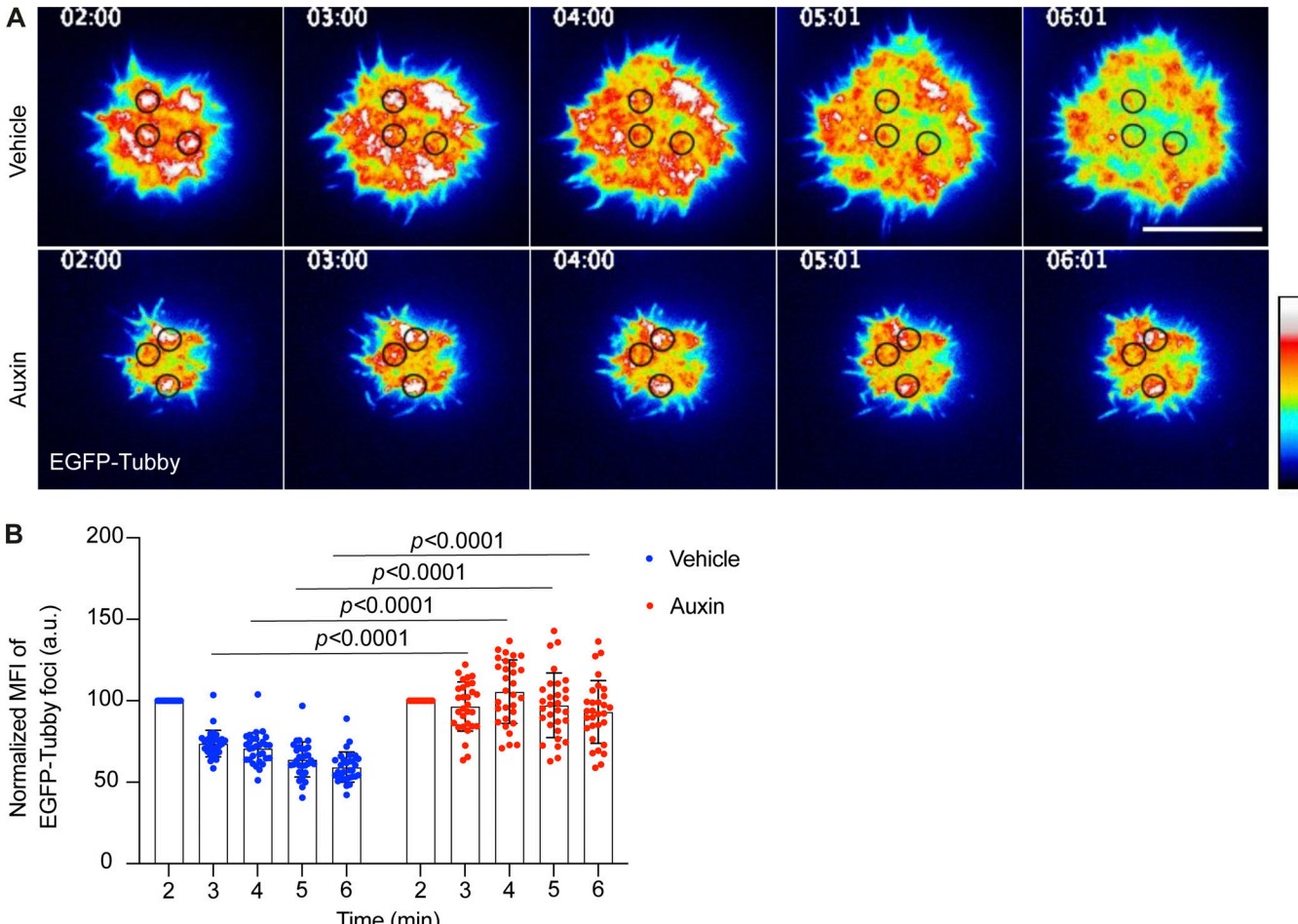JCB

**(C)** Representative confocal images at the indicated time points from auxin-treated INPP5B[Degron/WT] and INPP5B[Degron/G502D] cells stimulated with Texas Red–conjugated anti-IgM antibody. Scale bar, 10 µm. Statistical comparison for the percentage of cells showing BCR capping in INPP5B[Degron/WT] versus INPP5B[Degron/G502D] is shown on the right. Error bar represents SD, and the P value was calculated using Welch's *t* test. **(D)** Statistical comparison for the MFI of cell surface BCR from INPP5B[Degron/WT] vs. INPP5B[Degron/G502D] cells over time. Data were analyzed by two-way ANOVA, and P values were calculated using Sidak multiple comparisons test. Error bars indicate SD. **(E)** Representative TIRF microscopy images from auxin-treated INPP5B[Degron/WT] and INPP5B[Degron/G502D] cells that were settled on coverslips presenting surrogate antigens for 15 min. F-actin was stained with Phalloidin. Scale bar, 10 µm. **(F–H)** Statistical analyses of the TFI of BCR staining, spread area, and the MFI of BCR at the contact interface are shown in F, G, and H, respectively. Data (*n* = 35 cells) were pooled from three independent experiments. Bars represent mean ± SD, and P values were calculated using Welch's *t* test. **(I and J)** Protein extracts from auxin-treated INPP5B[Degron/WT] vs. INPP5B[Degron/G502D] cells stimulated in solution (I) or on glass coverslips (J) were blotted for Akt and ERK phosphorylation at the indicated time points. Data from three independent experiments were analyzed by two-way ANOVA, and P values were calculated using Sidak multiple comparisons test. Error bars indicate SEM. Source data are available for this figure: SourceData F3.

---

S3 D). This observation was validated by quantifying the decay of PI(4,5)P$_2$ within individual foci over time, which revealed that lipid turnover was indeed substantially reduced in INPP5B-depleted cells compared with control (Fig. 4 B). These findings suggest that INPP5B modulates PI(4,5)P$_2$ metabolism within discrete domains of the membrane contact area upon B cell activation.

We next wanted to investigate the spatial relationship between PI(4,5)P$_2$ turnover and BCR clustering, both in control cells and upon INPP5B depletion. PI(4,5)P$_2$ was therefore imaged simultaneously with BCR. In control cells, the removal of PI(4,5)P$_2$ occurred at sites of enrichment (foci) that subsequently became occupied by BCR microclusters (Fig. 5 A and Fig. 5, C–E). This finding suggests that PI(4,5)P$_2$ is removed from the regions

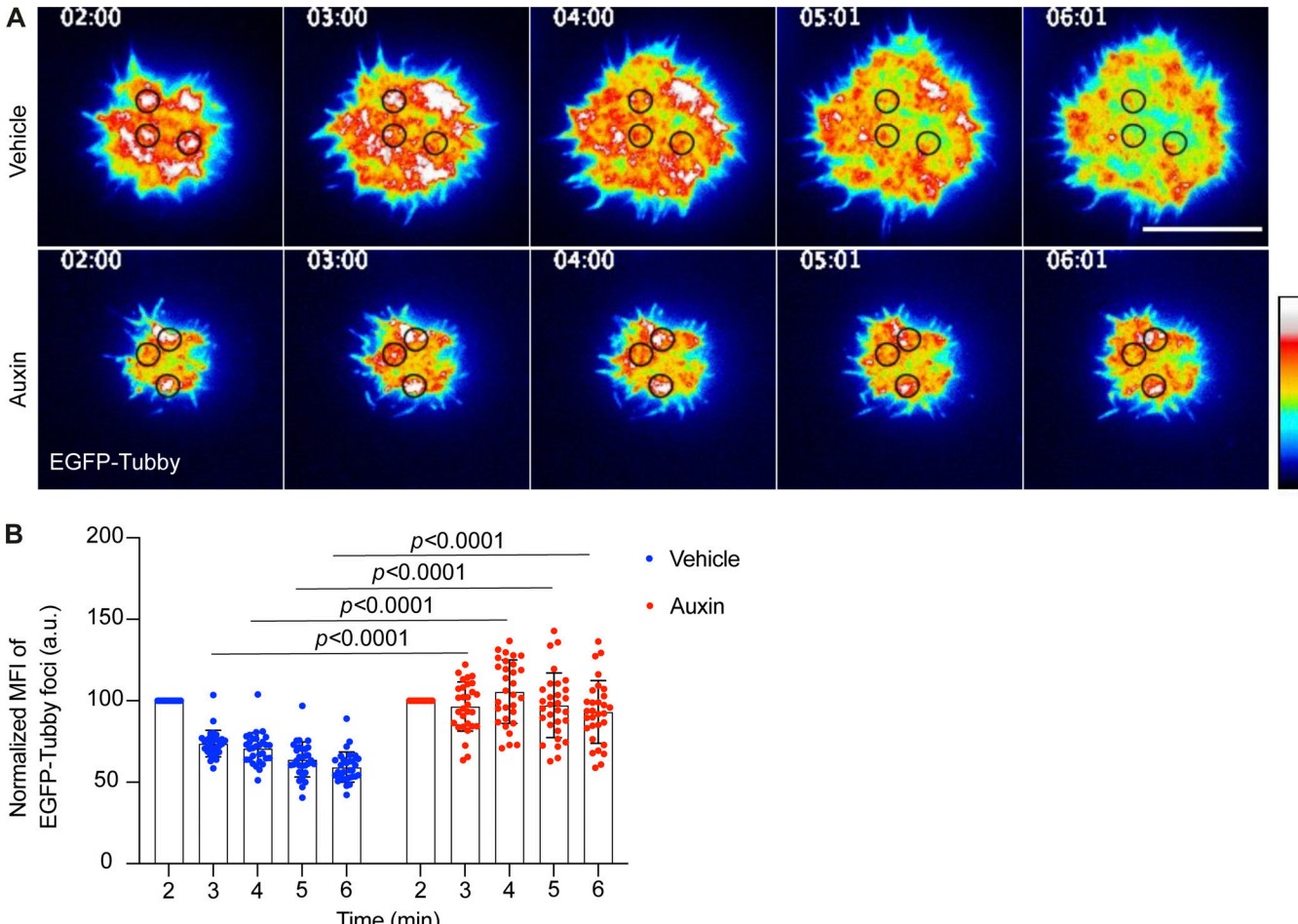

Figure 4. **PI(4,5)P$_2$ turnover within discrete domains of the membrane contact area is affected in INPP5B-depleted cells. (A)** Representative TIRF microscopy images at the indicated time points of INPP5B[Degron/Degron] cells expressing a fluorescent biosensor (tubby) after stimulation on antibody-coated coverslips. PI(4,5)P$_2$ regions of enrichment (foci) are denoted by black circles. Scale bar, 10 µm. See Videos 3 and 4 for the complete time-lapse TIRF microscopy images. **(B)** Statistical analysis of normalized MFI of the PI(4,5)P$_2$ biosensor in vehicle- vs. auxin-treated INPP5B[Degron/Degron] cells along time. Data from three independent experiments were analyzed by two-way ANOVA, and P values were calculated using Sidak multiple comparisons test. *n* = 30 foci; error bars indicate SD.

of the membrane where BCR clustering occurs, and that it might be a necessary prerequisite for this clustering to occur. As described above, the temporal dynamics of PI(4,5)P$_2$ turnover was disrupted in INPP5B-depleted cells (Fig. 5 B). The small amount of BCR clustering that was evident in these cells appeared to coincide with regions where a degree of PI(4,5)P$_2$ removal had occurred, and at a slower rate than in control cells (Fig. 5, C–E). Together, the results reveal a major role for INPP5B in mediating PI(4,5)P$_2$ removal that occurs prior to BCR clustering.

## INPP5B affects BCR clustering and signaling via control of cortical actin dynamics

PI(4,5)P$_2$ is a key regulator of actin dynamics, and it is well established that BCR clustering and cell spreading are highly dependent on dynamic actin remodeling (Li et al., 2018; Tolar, 2017). We therefore hypothesized that INPP5B exerts its effects on the BCR through control of actin dynamics. To examine this, we further manipulated the INPP5B$^{Degron/Degron}$ cells to stably express the F-actin probe LifeAct (Riedl et al., 2008) and used live TIRF microscopy to visualize F-actin in cells stimulated with coverslip-bound anti-IgM. As seen previously, rapid and sustained cell spreading occurred in control cells upon contact with the anti-IgM–coated coverslips (Fig. 6 A and Video 5). Cortical actin was dynamic during the attachment and cell spreading, as indicated by the rapid redistribution of the LifeAct probe. LifeAct initially accumulated at the contact interface as cells were landing on the coverslip and appeared in bright foci. As the cells began to spread, the staining at the initial contact area diminished, and the majority of foci disappeared, which was accompanied by a simultaneous enrichment of LifeAct labeling at the cell periphery (Fig. 6 A and Video 5), in a pattern similar to that observed for PI(4,5)P$_2$ (Fig. 4 A). In contrast, INPP5B-depleted cells exhibited dramatically impaired actin turnover, as indicated by the persistent and excessive accumulation of F-actin foci at the contact interface, as well as the lack of accumulation at the cell periphery, where spreading normally occurs (Fig. 6 A and Video 6). Quantitation revealed that control cells were able to clear ~50% of F-actin accumulated at the initial contact area over 5 min, whereas INPP5B-depleted cells showed no significant clearance of actin from the central region (Fig. 6 B). To determine the spatial relationship between actin and the BCR, they were imaged together. The results confirmed that the rapid clearance of F-actin from the central region correlates with BCR clustering in control cells (Fig. S4 A, top rows). However, in INPP5B-depleted cells, F-actin persisted at the contact interface and correlated with defective BCR clustering (Fig. S4 A, bottom rows). As can be seen in Fig. S4 B, the majority of F-actin staining in control cells disappeared (by 5 min) from the central region, where BCR clustering occurred, whereas no clearance of F-actin could be observed in INPP5B-depleted cells.

The ubiquitous requirement for continuous actin remodeling during B cell activation and spreading is orchestrated in large part by actin binding proteins (Li et al., 2018; Tolar, 2017) whose activities are altered by PI(4,5)P$_2$ (Janmey et al., 2018). To determine the mechanism by which INPP5B could be influencing actin dynamics, we analyzed the actin-severing protein cofilin (Freeman et al., 2011) and the ERM protein family member ezrin,

which links cortical actin filaments to the plasma membrane (Gupta et al., 2006; Treanor et al., 2011), both of which are controlled by PI(4,5)P$_2$. Actin severing by cofilin is inhibited by binding to PI(4,5)P$_2$ and is accompanied by dephosphorylation of Ser3 (Zhao et al., 2010), whereas ezrin actin–plasma membrane linking is promoted by PI(4,5)P$_2$ and associated with phosphorylation at Thr567 (Pelaseyed et al., 2017). In control cells, a low level of the inactive phosphorylated cofilin was present during starvation, and upon BCR stimulation, there was a slight increase in the level of phospho-cofilin (Fig. 6, C and D). Depletion of INPP5B resulted in higher phosphorylation of cofilin before BCR stimulation, which further increased upon stimulation, indicating less actin-severing activity by cofilin (Fig. 6, C and D). Ezrin phosphorylation was also increased in INPP5B-depleted cells compared with control following BCR stimulation, indicating increased ezrin cross-linking activity (Fig. 6, C and D). Thus, the ability of INPP5B to stabilize cortical actin may be explained, at least in part, by reduced cofilin and increased ezrin activity.

To further investigate how changes in cortical actin dynamics may be brought about in INPP5B-depleted cells, we analyzed the levels of activation of the small GTPases Rap1 and Cdc42. Both lie downstream of the BCR and are important for cortical actin reorganization in stimulated B cells. Rap1 plays a role in cofilin activation (Freeman et al., 2011), while Cdc42 has a number of roles, including activation of the LIM-kinase LIMK2 that can phosphorylate cofilin at Ser3 (Ohashi, 2015; Sumi et al., 1999). Using effector pulldown assays, we observed a comparable increase in Rap1 activation in both control and INPP5B-depleted cells, suggesting that INPP5B is not involved in the Rap1-cofilin axis (Fig. S4 C). In contrast, we observed a striking increase in Cdc42 activity upon INPP5B depletion, consistent with the increased cofilin phosphorylation we observed in INPP5B-depleted cells (Fig. 6 E).

Finally, it is well established that PI(4,5)P$_2$ is required for endocytosis (Posor et al., 2015). We therefore examined whether the reduced BCR clustering seen in INPP5B-depleted cells was simply a consequence of increased endocytosis due to increased PI(4,5)P$_2$ levels, as opposed to more direct effects on actin-dependent BCR clustering. INPP5B-depleted cells were treated with the dynamin inhibitor Dyngo-4a to inhibit endocytosis, and BCR capping with soluble antigen was examined as previously. BCR capping was unaffected by dynamin inhibition, even though it blocked BCR endocytosis (Fig. S5 A), indicating that -defective BCR clustering is independent of any effects of INPP5B depletion upon endocytosis. To further confirm that the effect of INPP5B depletion on capping is an actin-dependent phenomenon, we attempted to rescue the phenotype by reducing the extent of actin polymerization with low doses of Latrunculin A (LatA; Fujiwara et al., 2018). To this end, INPP5B-depleted cells were treated with 50 or 250 nM LatA before stimulation in solution. Treatment with 50 nM LatA fully restored BCR capping in INPP5B-depleted cells to control levels (Fig. S5 B), confirming that the phenotype is caused by excessive actin stabilization. Interestingly, the higher dose of LatA (250 nM) completely inhibited BCR capping and endocytosis in INPP5B-depleted cells, consistent with the requirement for actin in these processes (Fig. S5 B). Together, the findings reveal that INPP5B plays an

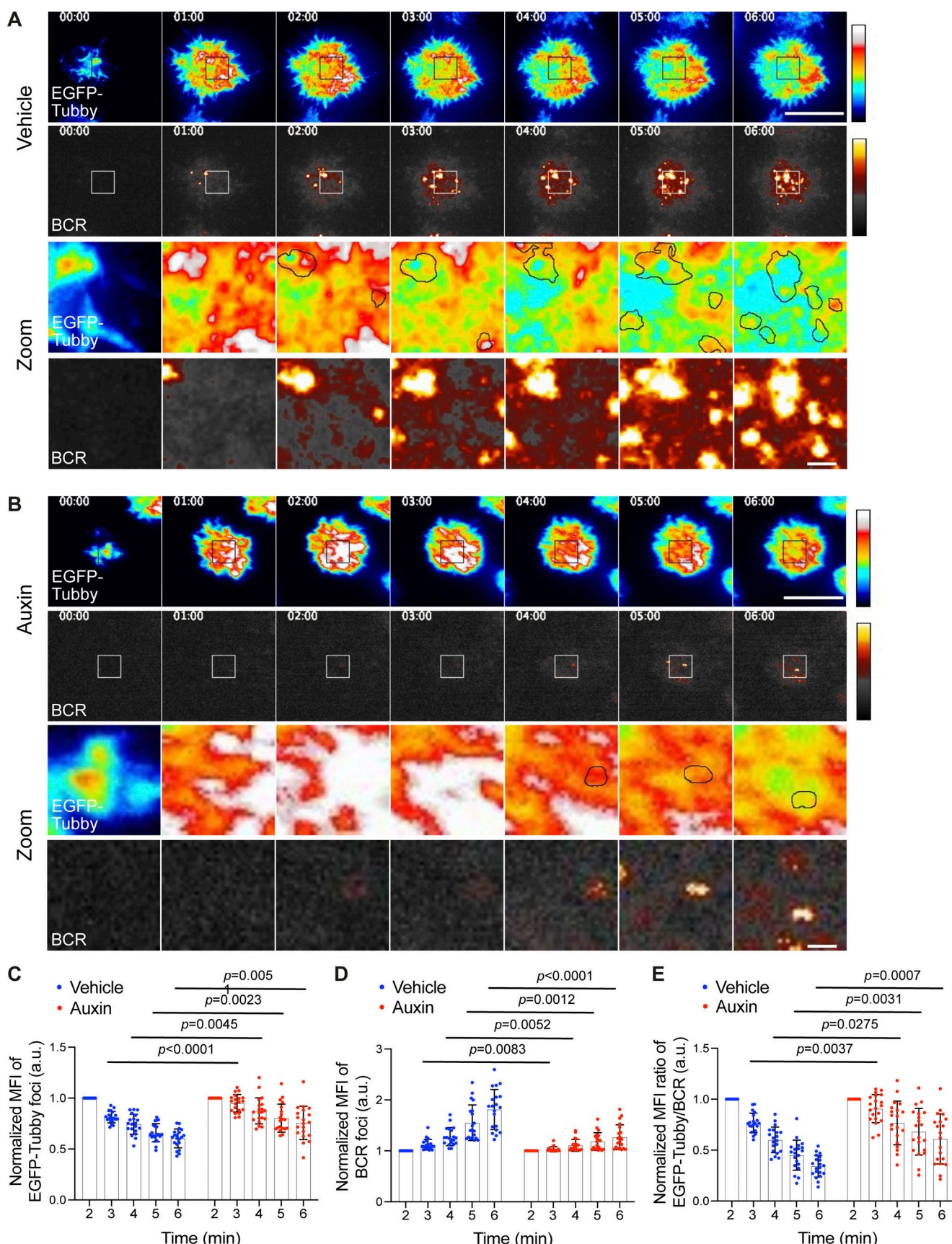

Figure 5. **Accumulation of PI(4,5)P$_2$ in INPP5B-depleted cells results in defective BCR clustering. (A)** Two-color time-lapse TIRF microscopy images of PI(4,5)P$_2$ (EGFP-Tubby) and BCR in vehicle-treated INPP5B$^{Degron/Degron}$ cells that were stimulated on antibody-coated glass. Images are pseudo-colored. Scale

bar, 10 µm. **(B)** Representative images of PI(4,5)P$_2$ and BCR in auxin-treated INPP5B$^{Degron/Degron}$ cells stimulated on glass as above. Boxed areas are magnified in time sequence (bottom rows). Scale bar, 1 µm. **(C)** Normalized MFI of PI(4,5)P$_2$ (EGFP-Tubby) foci over time starting 2 min after contact with the antibody-coated coverslip. n = 20 foci; error bars indicate SD. **(D)** Normalized MFI of BCR staining over time at the same membrane contact regions in C. **(E)** Normalized MFI ratio of PI(4,5)P$_2$/BCR over time. Data pooled from three independent experiments were analyzed by two-way ANOVA, and P values were calculated using Sidak multiple comparisons test. n = 20 foci (regions of PI(4,5)P$_2$ enrichment); error bars indicate SD.

important role in modulating dynamic actin organization in B cells, and that it does so through the actin-binding proteins cofilin and ezrin, to control BCR clustering and signaling.

## BCR-mediated apoptosis is reduced in INPP5B-depleted cells

B cells are continually guided by signals from the BCR and its co-receptors to make critical cell-fate decisions at different stages of their development, with responses varying depending on signal strength and complexity as well as the stage of B cell development (Niiro and Clark, 2002). For example, BCR ligation by foreign antigens on mature B cells results in pro-survival and proliferation to evoke a protective immune response against foreign antigens. However, BCR engagement on immature (pre-) B cells fails to initiate such mitogenic signals and instead results in cell death or a state of anergy (Koncz et al., 2002; Niiro and Clark, 2002). To determine whether reduced BCR signaling in INPP5B-depleted cells could impact BCR-mediated cell-fate decisions, we analyzed BCR-induced apoptosis as a functional measure of BCR signaling. DT40 cells are pre-B cells and therefore undergo apoptosis in response to BCR engagement in the absence of costimulatory signals (Takata et al., 1995). We therefore assessed apoptosis in BCR-stimulated INPP5B-depleted cells by annexin V staining (van Engeland et al., 1998). To this end, control and INPP5B-depleted cells were incubated in the presence of anti-IgM for 6 h, before labeling with annexin V, followed by flow cytometry analysis. Interestingly, INPP5B-depleted cells exhibited significantly weaker apoptotic response upon BCR stimulation (Fig. 7, A and B), which is consistent with an overall reduction in BCR signaling.

BCR-mediated apoptosis is dependent in part on the PLCγ$_2$-mediated calcium response (Tomlinson et al., 2001), which causes dephosphorylation and activation of the transcription factor NFATc1 to suppress activation-induced apoptosis in lymphocytes (Serfling et al., 2012). NFATc1 activation, assessed by phospho-antibody immunoblotting, was unaffected in INPP5B-depleted cells (Fig. 7, C and D), suggesting that the effect of INPP5B-depleted cells on BCR-mediated apoptosis is independent of NFATc1. We therefore also analyzed the FOXO1 transcription factor, which lies downstream of BCR signaling and upon phosphorylation prevents cell cycle arrest and/or apoptosis (Yusuf et al., 2004). FOXO1 phosphorylation at Ser256 was increased in INPP5B-depleted cells upon BCR stimulation (Fig. 7, C and D), which is consistent with the reduction in apoptotic response we observed in INPP5B-depleted cells. Overall, these data indicate that the impact of INPP5B on clustering and signaling of the BCR influences the physiological response of B cells to BCR stimulation.

## Cell spreading mediated by BCR signaling is reduced in human B cells lacking INPP5B activity

To determine whether the role of INPP5B in the B cell response to BCR stimulation is conserved between species, we analyzed two human B cell types: the well-studied Ramos B cell lymphoma model and primary B cells isolated from healthy donors. INPP5B was knocked out in Ramos cells using CRISPR/Cas9, which resulted in a complete loss of both INPP5B isoforms, the minor species at ~115 kD and the major form at ~75 kD, in both knockout (KO) clones that were generated (Fig. 8 A). We confirmed that the total cell surface BCR was unaltered in the KO clones (Fig. 8 B). To assess the effect of INPP5B loss on the BCR response, we assessed cell spreading on anti-IgM coated coverslips, which requires both BCR stimulation and dynamic actin remodeling downstream of BCR signaling. As shown in Fig. 8 C, upon activation, the WT Ramos cells spread effectively across the anti-IgM–coated coverslip, and spreading was significantly impaired in both INPP5B KO clones. We next used the chemical inhibitor YU142670, which inhibits the 5-phosphatase activity of INPP5B and the related phosphatase OCRL (Pirruccello et al., 2014). The YU inhibitor also significantly reduced cell spreading (Fig. 8 D). Importantly, this effect was lost in the INPP5B KO cells, confirming that the inhibitor is acting through inhibition of INPP5B (Fig. 8 D). We therefore used the YU inhibitor in primary human B cells, where again it significantly reduced the ability of cells to spread on anti-IgM–coated coverslips (Fig. 8 E). The magnitude of inhibition was comparable to that seen with an Arp2/3 inhibitor, previously shown to be important for cell spreading downstream of the BCR (Bolger-Munro et al., 2019), and slightly less than that seen upon chemical inhibition of Cdc42 (Gerasimcik et al., 2015; Fig. 8 F). Together these results confirm that the function of INPP5B in response to BCR stimulation is conserved in human B cells.

## Discussion

In this study we have identified the inositol 5-phosphatase INPP5B as a new player in the BCR response to simulation. Although it has been known for many years that the initial clustering of the BCR to form a signalosome is dependent on actin remodeling, how this remodeling is controlled remains poorly understood. Our work addresses this gap in knowledge by revealing a key role for INPP5B in this process. We show that INPP5B acts early following BCR stimulation to hydrolyze PI(4,5)P$_2$, which in turn promotes actin disassembly to allow receptor clustering and optimum downstream signaling. Our results are consistent with a model in which these effects are mediated through activation and inactivation of the actin-severing protein cofilin and actin-membrane linker ezrin, respectively. Considering that PI(4,5)P$_2$ can regulate a number of actin regulators (Janmey et al., 2018), we would speculate that INPP5B action upon the actin cytoskeleton in this context is likely to be mediated by additional factors. Indeed, we also observed an increase in the level of active Cdc42 in INPP5B-depleted cells,

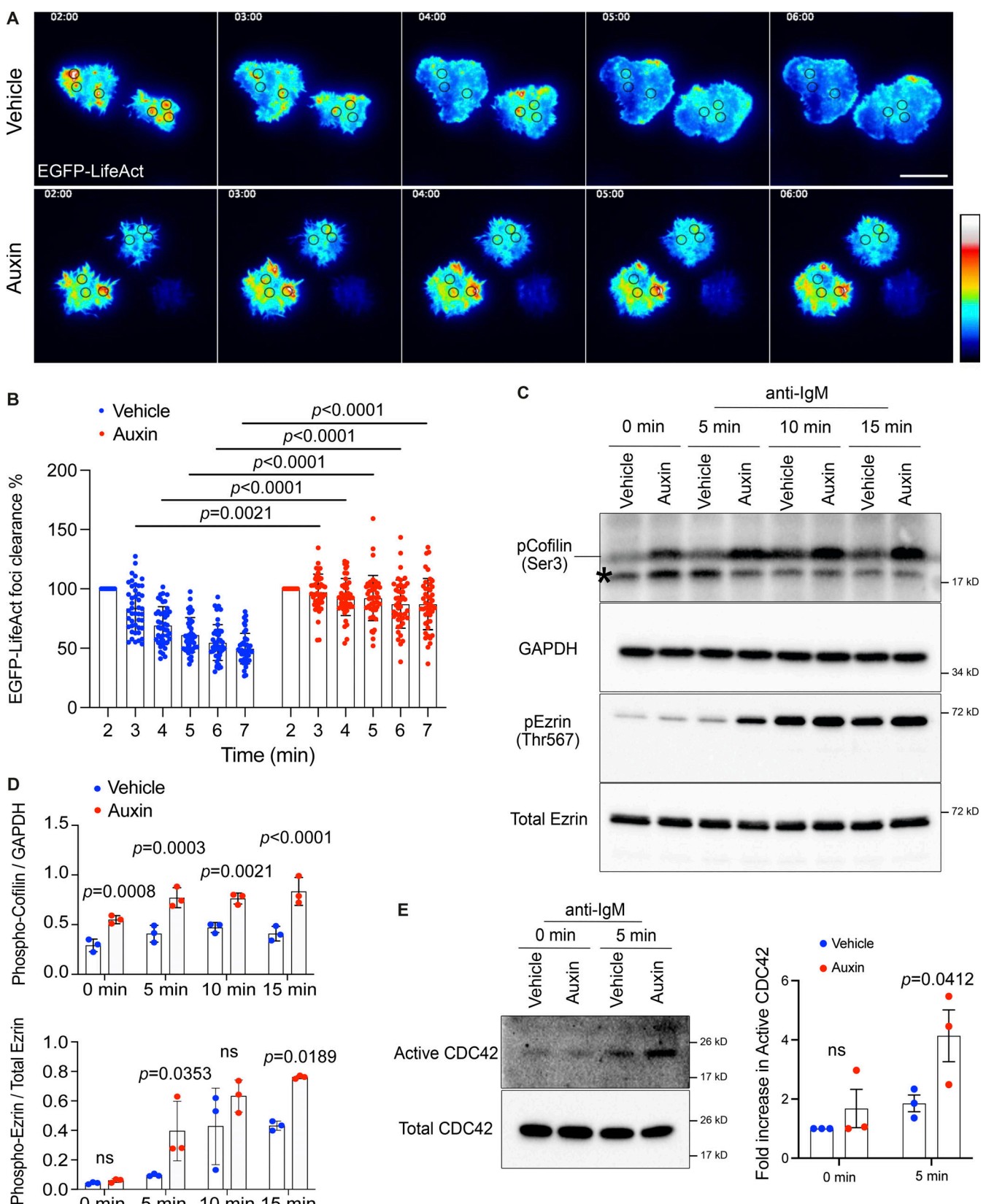

Figure 6. **The dynamics of cortical actin is altered in INPP5B-depleted cells. (A)** Representative TIRF microscopy images at the indicated time points of INPP5B[Degron/Degron] cells expressing LifeAct-EGFP after stimulation on immobilized anti-IgM. See Videos 5 and 6 for the complete time-lapse TIRF microscopy images. **(B)** Analysis of F-actin (LifeAct-EGFP) foci clearance along time in vehicle- vs. auxin- treated INPP5B[Degron/Degron] cells. Data (*n* = 45 foci) from three independent experiments were analyzed by two-way ANOVA, and P values were calculated using Sidak multiple comparisons test. Error bars represent

mean ± SD. **(C)** Protein extracts from BCR-stimulated INPP5B[Degron/Degron] cells were blotted for cofilin and ezrin phosphorylation at the indicated time points. The asterisk marks a nonspecific band. **(D)** Quantification of three such blots by densitometry is shown in D for cofilin (top) and ezrin (bottom). Bars represent mean ± SD, and P values were calculated as above. **(E)** GTP-bound Cdc42 was detected using GST-PAK1 RBD pulldown followed by blotting against Cdc42. Quantification of three such blots by densitometry is shown on the right. GTP levels of Cdc42 are expressed relative to total Ras, and data were normalized to unstimulated vehicle (control). Data were analyzed by two-way ANOVA, and P values were calculated using Sidak multiple comparisons test. Error bars indicate SEM. Source data are available for this figure: SourceData F6.

which could be mediated by the PI(4,5)P₂-binding Rho family GEFs Vav (Treanor et al., 2011; Weber et al., 2008) or DOCK8 (Sakurai et al., 2021). Similarly, the actin nucleation promoting factor WASP (Huang et al., 2020; Rey-Suarez et al., 2020), which is regulated by PI(4,5)P₂ and Cdc42 (Rohatgi et al., 2000), may also be involved. Clearly, though, cofilin and ezrin are critical regulators of BCR mobilization that are involved early in the BCR response (Freeman et al., 2011; Gupta et al., 2006; Hao and August, 2005; Treanor et al., 2011) and represent major downstream targets of INPP5B in this process.

The involvement of INPP5B in BCR clustering extends beyond removal of the cortical network to promote lateral BCR diffusion at the plasma membrane. The spreading of B cells on surface antigen is an important step in the acquisition and gathering of antigen-bound BCR into a central cluster at the immune synapse (Batista et al., 2001; Fleire et al., 2006). Loss of

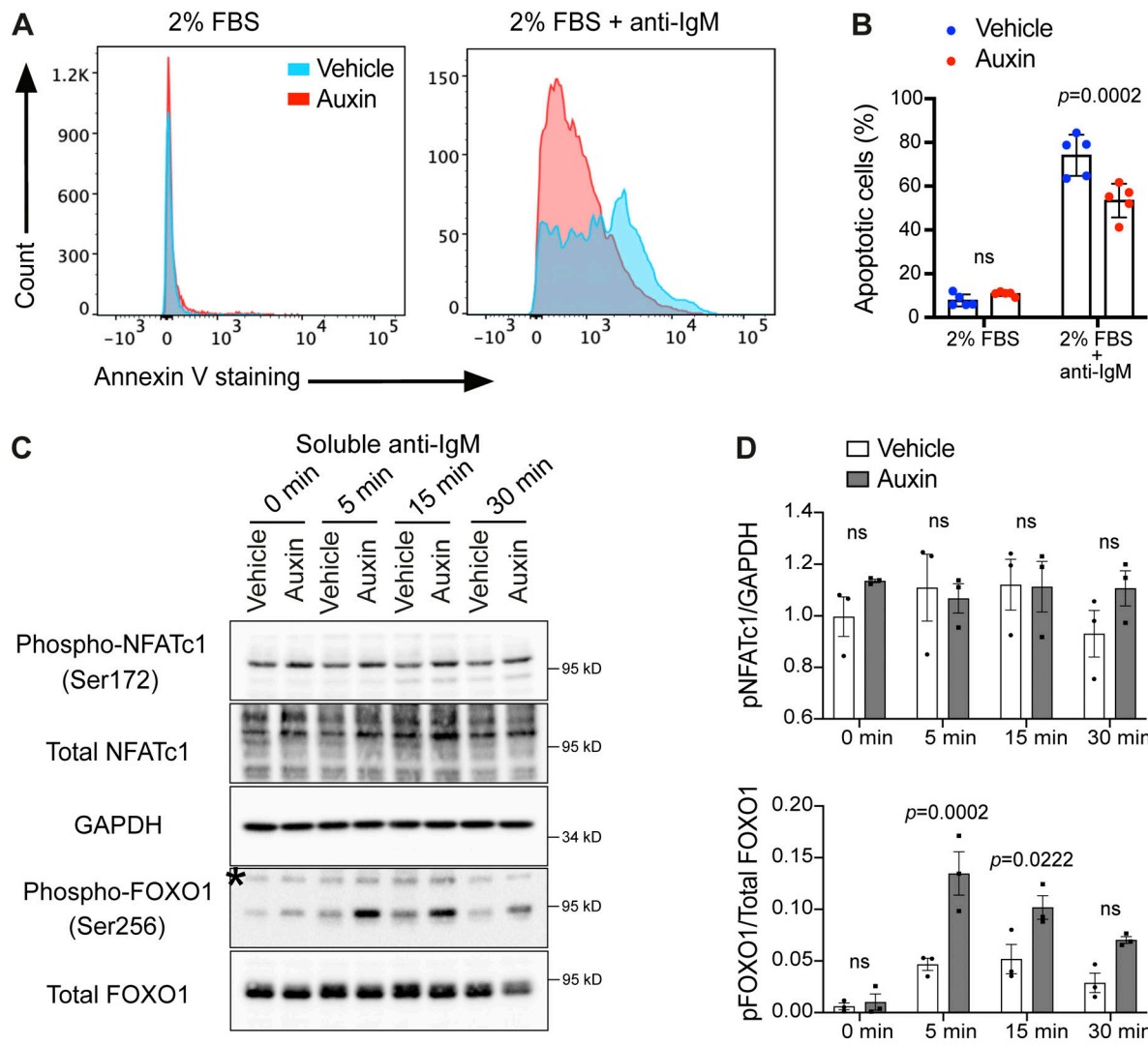

Figure 7. **BCR-mediated apoptosis is reduced in INPP5B-depleted cells. (A)** Representative histograms showing the intensities of FITC-annexin V staining of INPP5B-degron cells that were cultured in RPMI-1640 containing 2% FBS with or without anti-IgM for 6 h. **(B)** Apoptotic cells are expressed as a percentage of cells showing annexin V staining. Data were pooled for early (PI-negative) and late (PI-positive) apoptosis and analyzed by two-way ANOVA. **(C)** Protein extracts from BCR-stimulated INPP5B[Degron/Degron] cells were blotted for NFATc1 and FOXO1 phosphorylation at the indicated time points. **(D)** Quantification of three such blots by densitometry. Error bars represent SEM. Data were analyzed by two-way ANOVA, and P values were calculated using Sidak multiple comparisons test. Source data are available for this figure: SourceData F7.

Figure 8. **BCR-mediated spreading is reduced in human B cells lacking INPP5B. (A)** Protein extracts from WT Ramos and INPP5B[KO/KO] cells were subjected to IP using sheep anti-INPP5B antibody. INPP5B was detected by Western blotting using a rabbit anti-INPP5B. **(B)** Analysis of BCR surface expression in WT vs. INPP5B-KO clones by flow cytometry. **(C)** Representative TIRF microscopy images from WT vs. INPP5B-KO cells that were settled on coverslips presenting surrogate antigens for the indicated time points. F-actin was stained with Phalloidin. Scale bar, 10 µm. Statistical analyses of cell spreading area (at 5 and 15 min) are shown on the right. The data are pooled from three independent experiments. Error bars indicate SD, and P values were calculated using Tukey's multiple comparisons test. **(D)** Representative TIRF microscopy images from DMSO (vehicle)- vs. INPP5B inhibitor (YU142670)–treated WT and INPP5B-KO clone 2 cells settled on coverslips presenting surrogate antigens. Analysis of cell spreading was performed as in C, and data on the right were pooled from three independent experiments. **(E)** Freshly isolated untouched human primary B cells were treated with DMSO (vehicle) vs. INPP5B inhibitor for 30 min, before activation on coverslips presenting surrogate antigens. Cell spreading was analyzed at 15 min, and data on the right were pooled from four independent experiments (using PBMCs isolated from three different healthy donors). The P value was calculated using Wilcoxon matched-pairs signed rank test. **(F)** Human primary B cells treated with DMSO vs. Cdc42 inhibitor (ML141) vs. Arp2/3 inhibitor (CK666) were settled on antibody-coated coverslips and analyzed as previously. The data represent mean ± SD of 45 cells, and P values were calculated using Dunnett's T3 multiple comparisons test. Source data are available for this figure: SourceData F8.

INPP5B dramatically impaired cell spreading, which is also likely to contribute to the reduced BCR clustering that was observed. This effect is also due to a failure to hydrolyze PI(4,5)P$_2$, and again cofilin and additional PI(4,5)P$_2$-binding regulatory factors are likely to be involved (Bolger-Munro et al., 2021; Freeman et al., 2011; Li et al., 2018; Tolar, 2017). Cell spreading involves constant spatially controlled turnover of the actin network, and these factors are required to undergo rounds of

activation and inactivation to allow actin remodeling to occur, driven by constant $PI(4,5)P_2$ turnover. It is therefore likely that INPP5B is required throughout the spreading process to maintain actin remodeling. It is interesting to note that the related 5-phosphatase OCRL has been implicated in regulating levels of active Rac and Rho in other cell types (Egot et al., 2021; van Rahden et al., 2012), and it will be interesting to investigate this possibility in the context of B cells, especially considering that Vav is also a Rac GEF, and Rac, which promotes B cell spreading, is activated downstream of the BCR (Arana et al., 2008; Weber et al., 2008).

It was recently reported that $PLC\gamma_2$ mediates $PI(4,5)P_2$ hydrolysis in BCR microclusters upon receptor stimulation, and that this hydrolysis is required for microcluster growth (Wang et al., 2017; Xu et al., 2017). Previous work also suggested a role for $PLC\gamma_2$ in regulating ezrin activity downstream of the BCR (Treanor et al., 2011). How does this fit with a role for INPP5B in BCR clustering? Our data suggest an early role for INPP5B in removal of $PI(4,5)P_2$ to promote BCR clustering. We envisage this as being upstream from the involvement of $PLC\gamma_2$, which is found within the clusters (Xu et al., 2017). This could be mediated by early recruitment of INPP5B to BCR microclusters, or it is even possible that INPP5B may have a constitutive role in regulating cortical actin via $PI(4,5)P_2$, which could be important for tonic signaling as well as providing a more permissive environment for BCR clustering upon antigen binding. Indeed, our results with phospho-cofilin and Cdc42 suggest such a role. An alternative possibility is that INPP5B and $PLC\gamma_2$ function in tandem within clusters to hydrolyze $PI(4,5)P_2$, but if this were the case the dramatic effects we observed upon loss of INPP5B would suggest it is the dominant enzyme. The products of $PLC\gamma_2$-mediated $PI(4,5)P_2$ hydrolysis recruit PIPK for resynthesis of $PI(4,5)P_2$ adjacent to the BCR clusters, and this amplification step helps drive cluster growth (Xu et al., 2017). However, INPP5B, even if present within clusters, would not be expected to contribute to this process, as it generates a different product from $PLC\gamma_2$, namely PI4P as opposed to DAG and $IP_3$. Of interest, $PI(3,4,5)P_3$ has been shown to function in the growth of BCR clusters by promoting the recruitment of the Rac GEF DOCK2 (Wang et al., 2017). PI3K and PTEN activity within BCR clusters maintains an equilibrium between $PI(4,5)P_2$ and $PI(3,4,5)P_3$ levels, allowing for optimal recruitment of DOCK2 and control of actin dynamics and cluster growth. Interestingly, INPP5B-depleted cells have reduced $PI(3,4,5)P_3$ levels due to attenuated PI3K signaling, and hence INPP5B could also impact cluster growth via DOCK2 in an indirect manner, as a consequence of reduced signalosome formation that lies downstream of impaired BCR clustering.

Previous studies have revealed apparent functional redundancy between OCRL and INPP5B, at least in the context of mouse development (Janne et al., 1998) and ciliogenesis (Luo et al., 2013). This would fit with the proteins having a shared number of interaction partners including Rab GTPases, APPL1, IPIP27A and B, and Rac/Cdc42 (De Matteis et al., 2017; Mehta et al., 2014). However, the degree of functional redundancy between these two 5-phosphatases remains poorly defined. In the context of B-lymphocytes, there is no functional requirement for OCRL in BCR clustering and signaling, whereas INPP5B is clearly important. This can be explained by the proteins having different functional roles within lymphocytes, since both OCRL and INPP5B are expressed in this cell type. Additional interacting partners that are unique to INPP5B may be relevant here, or differential regulation of INPP5B, perhaps downstream of the BCR, may be important. Further work will be required to distinguish the unique features of INPP5B that make it so important for BCR dynamics. It is also worth pointing out that although both enzymes are expressed in lymphocytes, INPP5B is expressed at higher levels than OCRL, and this is also likely relevant for the cell-type dependence on INPP5B we observed (https://www.gtexportal.org/home/).

Excessive or altered BCR signaling is a major determinant in B cell cancers such as lymphoma and lymphocytic leukemia (Burger and Wiestner, 2018; Young and Staudt, 2013), as well as in autoimmune disorders (Rawlings et al., 2017). Consequently, various drugs have been developed to target this pathway, with inhibitors of PI3Kδ and the tyrosine kinase BTK approved for use in various types of B cell cancers (Burger and Wiestner, 2018). Despite their obvious benefits, both drugs, which are kinase inhibitors, vary in their efficacy, and resistance to therapy commonly occurs (Klener and Klanova, 2020). Hence, there remains a need for new drugs and the identification of novel drug targets that could be exploited in a therapeutic context. INPP5B is an enzyme that is "druggable," and targeting this enzyme is unlikely to produce severe side effects, as the KO mouse model developed normally and was viable (Janne et al., 1998). It is worth noting that the INPP5B KO mouse has not been studied in terms of the immune system, which would certainly be interesting in light of our current study. Because INPP5B acts through the actin cytoskeleton to control BCR clustering and signaling, it differs from the other drugs currently in the clinic or undergoing trials, which target the downstream signaling axis (Efremov et al., 2020). Hence, it is a promising candidate target for drugs to alleviate B cell malignancies attributable to dysregulated BCR signaling.

## Materials and methods
### Materials and reagents
All materials, reagents, and antibodies used in this study are listed in Table S1 or mentioned in the text.

### Cell culture
DT40 cells were a kind gift of Prof. R.F. Irvine (retired, University of Cambridge, Cambridge, UK). Cells were cultured in RPMI-1640 supplemented with 10% FBS, 2% chicken serum, 2 mM glutamine, 50 μM 2-mercaptoethanol, 50 U/ml penicillin, and 50 μg/ml streptomycin. Cells were cultured at 41°C in 5% $CO_2$ and maintained at a density of 2–4 million cells per milliliter of growth medium. Auxin stock was prepared fresh in distilled water (vehicle) and diluted in culture medium to a final concentration of 500 μM. Ramos B cells were obtained from ATCC (CRL-1596). Cells were cultured in RPMI-1640 supplemented with 10% FBS, 2 mM glutamine, 50 U/ml penicillin, and 50 μg/ml streptomycin. Cells were cultured at 37°C in 5% $CO_2$ and were

maintained at a density of 0.2–1.5 million cells per milliliter of growth medium. Primary B cells were isolated from frozen human peripheral blood mononuclear cells (PBMCs; healthy donors) using EasySep Human B cell Isolation Kit (STEMCELL Technologies) following the manufacturer's instructions. Freshly isolated B cells were used in experiments on the same day.

### Generation of the targeting constructs to endogenously tag *INPP5B* at the C-terminus with the auxin degron

The 2.3-kb 5′ homology arm of the *INPP5B* tagging construct was amplified by PCR from DT40 genomic DNA immediately upstream of *INPP5B* stop codon using primers 5′ARM-INPP5B-F/R and cloned into pBluescript SK+ between ClaI and EcoR1 to yield pINPP5B-AID-01. The auxin degron tag comprising a linker followed by AtIAA17-FLAG-$(His_6)_2$ was PCR-amplified from pAAIDII-5 plasmid (Bulley et al., 2016) using primers PCR-Degron-F/R and cloned into pINPP5B-AID-01 between EcoR1 and BamH1 to yield pINPP5B-AID-02. The 2.3-kb 3′ homology arm of the INPP5B tagging construct was amplified by PCR from DT40 genomic DNA immediately downstream of *INPP5B* stop codon using primers 3′ARM-INPP5B-F/R and cloned into pBluescript SK+ between Kpn1 and XbaI to yield p3′-INPP5B-01 plasmid. An unwanted BamH1 site within the 3′ arm was removed by site-directed mutagenesis (SDM) using primers SDM-INPP5B-F/R. The 3′ arm was then amplified by PCR using primers PCR-m3′-INPP5B-F/R and cloned into pINPP5B-AID-02 between BamH1 and Not1 to yield pINPP5B-AID-03. Finally, blasticidin and puromycin resistance cassettes (Arakawa et al., 2001) were cloned into pINPP5B-AID-03 using the BamH1 restriction site to yield pINPP5B-AID-04 and pINPP5B-AID-05, respectively.

### Targeted transfection in DT40 cells

Targeting constructs were linearized using an appropriate restriction enzyme and recovered by lithium chloride precipitation. $20 \times 10^6$ cells were harvested during the logarithmic phase of growth and washed once in growth medium. Cells were then resuspended and mixed with the DNA solution in an electroporation cuvette with a 4-mm electrode gap. The cells were then transfected using an exponentially decaying electrical pulse (600 V, 25 μF; Bio-Rad Gene Pulser Xcell). Next, the cells were immediately diluted in growth medium and allowed to recover overnight. On the next day, the appropriate selection antibiotic was added, and cells were seeded in 96-well plates. Cells were incubated for 7 d, by which time single antibiotic-resistant colonies were visible. Resistant colonies were picked from wells containing only one colony and were subcultured for further analysis. gDNA was harvested to perform test PCR and sequencing reactions to check for successful construct targeted integration.

### Generation of the INPP5B[Degron/Degron] cells

pINPP5B-AID-04 plasmid was linearized by overnight digestion with Not1 and purified. Transfection of a DT40 cell line stably expressing osTIR1 (R.F. Irvine) with the linearized plasmid was achieved as described above. To select for stable clones, blasticidin was used at a final concentration of 20 μg/ml. Resistant clones were screened for targeted integration of pINPP5B-AID-04 sequence using primers SCR-pINPP5B-AID-F, which anneals

immediately upstream of the 5′ homology arm, and SCR-AMD1-R, which anneals within the IAA17 (degron) sequence. A product of 2.7 kb was a result of successful targeted integration. To further edit the second allele of INPP5B in INPP5B[Degron/WT] cells, a second round of targeted transfection with pINPP5B-AID-05 was conducted. Successful transfection was screened by selection with blasticidin and puromycin. Dual-resistant clones were screened for successful tagging using primers Test-INPP5B-F and Test-INPP5B-R, which flank a 680-bp site near the last exon of INPP5B (the site of integration). Failing to amplify through the genomic locus (due to the presence of a chicken β-actin promoter; Arakawa et al., 2001) suggested successful tagging of both alleles of INPP5B.

### Cre/loxP-mediated excision of antibiotic resistance cassettes

Cells were transfected with pLV-EGFP-Cre (#86805; Addgene). 24 h later, cells were FACS-sorted based on the EGFP signal. Successful excision of antibiotic resistance genes was verified by PCR on gDNA and by culturing cells in selective medium.

### Generation of the targeting constructs to endogenously tag *OCRL* at the C-terminus with the auxin degron

The 2.4-kb 5′ homology arm of the *OCRL* tagging construct was amplified by PCR from DT40 genomic DNA immediately upstream of *OCRL* stop codon using primers 3′ARM-OCRL-F/R and cloned into pBluescript SK+ between Sal1 and EcoR1 to yield pOCRL-AID-01. The auxin degron tag was PCR-amplified as above and cloned into pOCRL-AID-01 between EcoR1 and BamH1 to yield pOCRL-AID-02. The 2.8-kb 3′ homology arm of the *OCRL* tagging construct was then amplified by PCR from DT40 genomic DNA immediately downstream of *OCRL* stop codon using primers 3′ARM-OCRL-F/R and cloned into pBluescript SK+ between Sal1 and EcoR1 to yield p3′-OCRL-01 plasmid. An unwanted BamH1 site within the 3′ arm was removed by SDM using primers SDM-OCRL-F/R. The 3′ arm was then amplified by PCR using primers PCR-m3′-OCRL-F/R and cloned into pOCRL-AID-02 between BamH1 and Not1 to yield pOCRL-AID-03. Finally, blasticidin and puromycin resistance cassettes were cloned into pOCRL-AID-03 using the BamH1 restriction site to yield pOCRL-AID-04 and pOCRL-AID-05, respectively.

### Generation of the targeting construct to introduce a G502D point mutation in *INPP5B*

The 1.9-kb 3′ homology arm of the *INPP5B* targeting construct was amplified by PCR from DT40 genomic DNA using primers 3′-INPP5B-KI-F/R and cloned into pBluescript SK+ between BamH1 and Xba1 to yield pINPP5B-KI-01. The 2.55-kb 5′ homology arm of the *INPP5B* targeting construct was amplified by PCR from DT40 genomic DNA using primers 5′-INPP5B-KI-F/R-01 and cloned into pBluescript SK+ between Sal1 and ClaI. An unwanted BamH1 site within the 5′ arm sequence was removed by SDM using primers SDM-INPP5B-1-F/R, before introducing a D502G mutation using primers SDM-INPP5B-2-F/R. The 5′ arm was then PCR-amplified using primers 5′-INPP5B-KI-F/R-02 and cloned into pINPP5B-KI-01 to yield pINPP5B-KI-02. Finally, a puromycin resistance cassette was cloned into pINPP5B-KI-02 using the BamH1 restriction site to yield pINPP5B-KI-03.

## Generation of the INPP5B$^{Degron/G502D}$ cells

INPP5B$^{Degron/WT}$ cells were transfected with pINPP5B-KI-03 plasmid as described above, and stable transfectants were selected using blasticidin and puromycin treatments. To screen for successful targeted mutagenesis of the degron-untagged allele, a two-step strategy based on genomic DNA analysis by PCR was used. The first step aimed at examining the correct integration of the targeting construct into the untagged allele using primers SCR-SDM-Integ-F/R. A product of 2 kb was a result of successful targeted integration into the degron-untagged allele. The second step aimed at examining the success of introducing the point mutation and involved the PCR amplification of a 2.5-kb fragment from potential clones using primers SCR-SDM-KI-F/R, followed by DNA sequencing.

## Stable nontargeted transfection of DT40 cells

Stable nontargeted transfectants were generated and expanded from single-cell clones in exactly the same way as targeted transfectants, except for the electrical pulse, which was delivered at 250 V, 950 µF. Correct expressions were verified by Western blotting and flow cytometry. To generate INPP5B$^{Degron/Degron}$ cells stably expressing LifeAct-EGFP and EGFP-Tubby, cells were transfected with pcDNA-Hygro-LifeAct and pcDNA-Hygro-Tubby, respectively. Stable clones were selected by Hygromycin B at a final concentration of 2 mg/ml in culture medium for 5 d, before FACS-sorting expressing cells based on the EGFP signal. To generate the expressing plasmids, the insert in pcDNA3.1-Hygro-EYFP-H148Q plasmid (#25873; Addgene) was replaced with the coding sequences for EGFP-Tubby or LifeAct-EGFP to yield pcDNA-Hygro-Tubby or pcDNA-Hygro-LifeAct, respectively. The coding sequences for EGFP-Tubby and LifeAct-EGFP were PCR-amplified from pEGFP-C1-Tubby (Martin Lowe laboratory, University of Manchester) and pEGFP-C1-LifeAct-EGFP (#58470; Addgene) using primers PCR-EGFP-Tubby-F/R and PCR-Lifeact-EGFP-F/R, respectively.

## Editing Ramos cells with CRISPR/Cas9

gRNAs targeting *INPP5B* were designed using the CRISPOR website (http://crispor.tefor.net) along with the IDT CRISPR-Cas9 gRNA checker (https://eu.idtdna.com/site/order/designtool/index/CRISPR_SEQUENCE) to identify the most efficient and specific guides. Selected Alt-R custom sgRNAs were ordered from IDT. The gRNA sequence used to generate the INPP5B KO line was 5′-CUUUCACGAUACCCCAAAGG-3′. Before transfection, 200 pmol of Cas9 protein (IDT) was complexed with 400 pmol of sgRNA (IDT) using the Lonza SG nucleofection kit (V4XC-3024; Lonza) for 20 min at room temperature. $2 \times 10^6$ Ramos B cells were resuspended in the Cas9/sgRNA mixture and electroporated with program CA-137 using a 4D-Nucleofector X Unit (Lonza). Cells were allowed to recover in complete culture medium for >7 d before sorting by FACS. Single-cell clones were allowed to expand before downstream analysis. Genomic DNA was extracted from single-cell clones and the region targeted by gRNAs was amplified using PCR. INPP5B KO was confirmed at the genomic level using the Inference of Crispr Edits online tool (Synthego; https://ice.synthego.com) and at the protein level via IP with in-house antibodies raised against INPP5B (Williams et al., 2007).

## Preparation of cell-free extracts

DT40 cells were lysed in ice-cold lysis buffer (50 mM Tris-HCl, pH 7.4, 150 mM NaCl, 1 mM EDTA, 1% Triton X-100, 1% v/v protease inhibitor cocktail, and 1% v/v phenylmethanesulfonyl fluoride solution [0.1 M in ethanol]) and incubated on ice for 20 min. The lysate was cleared by centrifugation at 16,000 $g$ for 30 min at 4°C to yield the protein extract. Protein concentration was measured using the Bradford protein assay, and lysates were normalized to the lowest concentration.

## Immunodetection of INPP5B

Degron-tagged INPP5B was immunoprecipitated by using its FLAG tag. ANTI-FLAG M2 Affinity Gel (a mouse FLAG monoclonal antibody covalently attached to agarose) was used. Cell-free protein extracts were prepared as described above and were precleared by incubation with 30 µl agarose beads for 30 min at 4°C with gentle agitation. The agarose beads were then pelleted by centrifugation at 13,000 $g$ for 1 min, and the supernatant was transferred into a new tube. IP of FLAG-tagged proteins was achieved by incubation of the precleared protein extracts with 20 µl of M2 Affinity resin (before use, storing glycerol was discarded, and the resin was washed twice in ice-cold lysis buffer) with constant rotation for 2 h at 4°C. Beads were then pelleted by centrifugation at 13,000 $g$ for 1 min and washed three times in ice-cold PBS containing 0.1% Triton X-100 and then twice in ice-cold PBS (with no detergent). Beads were then mixed (1:1 v/v) with 2× SDS loading buffer and boiled for 5 min to elute proteins. IP products were separated by 8% SDS-PAGE and used for Western blot with anti-His antibody. To deplete INPP5B, indole-3-acetic acid sodium salt was added to the medium at a final concentration of 500 µM. To detect INPP5B in Ramos cells, the protein was immunoprecipitated using an in-house sheep anti-INPP5B antibody, followed by immunoblotting with an in-house rabbit polyclonal anti-INPP5B antibody (Williams et al., 2007).

## Western blotting

Protein extracts were denatured by boiling in 1:1 v/v 2× SDS loading buffer (100 mM Tris-HCl, pH 6.8, 4% w/v SDS, 0.2% w/v bromophenol blue [Sigma-Aldrich], 20% v/v glycerol, and 200 mM 2-mercaptoethanol) for 5 min. Proteins were separated by SDS-PAGE and transferred to nitrocellulose membranes. Membranes were blocked with 5% w/v nonfat milk for 1 h at room temperature before incubation with primary antibodies at 4°C overnight (or 2 h at room temperature). HRP-conjugated secondary antibodies were used to detect and visualize the target proteins. Western blots were developed with the Bio-Rad ChemiDoc Imaging System, using the auto-exposure setting, and analyzed with Bio-Rad Image Lab 6.1 software.

## Spinning disc confocal live imaging

35-mm Ibidi imaging dishes with glass bottoms were coated with poly-L-lysine (0.01% solution) for 30 min at room temperature. $2 \times 10^6$ cells per dish were plated in starvation medium (with or without auxin) for 2 h at 41°C. Dishes were then placed on ice, and the starvation medium was replaced with ice-cold starvation medium containing 1 µg/ml Texas Red–conjugated

polyclonal anti-chicken IgM antibody. BCR labeling was achieved on ice for 30 min. Cells were then washed once with ice-cold PBS and left on ice in starvation medium containing 20 mM Hepes until imaging. Prewarmed starvation medium was added to cells immediately before image acquisition. 50 time points were acquired comprising a total of 10 confocal sections each, with a step size of 0.6 µm (a total range of 5.4 µm). Exposure time was set to 200 ms for the BCR. To stain acidic compartments, LysoTracker Green DND-26 was used at a final concentration of 20 nM. For rescue experiments, compounds were added to the cells for the last 30 min of starvation and kept in the imaging medium. Images were acquired using a CSU-X1 spinning disc confocal assembly (Yokagawa) on a Zeiss Axio-Observer Z1 microscope with a 60×/1.40 Plan-Apochromat objective, Evolve EMCCD camera (Photometrics), and motorized XYZ stage (ASI). The 488- and 561-nm lasers were controlled using an acousto-optic tunable filter through the laser stack (Intelligent Imaging Innovations [3i]) allowing both rapid "shuttering" of the laser and attenuation of the laser power. Slidebook software (3i) was used to capture images. Live imaging was performed at 39.5–41°C.

## Analysis of BCR endocytosis by flow cytometry

$40 \times 10^6$ cells were harvested during the logarithmic growth phase and washed once in PBS before incubation in starvation medium (RPMI-1640, 1% BSA, and L-glutamine) for 2 h. At the end of incubation time, cells were collected, the resulting pellet was resuspended in 1 ml ice-cold starvation medium containing 1 µg mouse anti-chicken IgM antibody, and cells were incubated for 30 min with frequent pipetting. At the end of labeling, cells were harvested and resuspended in 800 µl ice-cold PBS and split into 200 µl (first) and 600 µl (second). Cells were collected again by centrifugation. As a control, the first pellet was resuspended in 200 µl ice-cold starvation medium and placed on ice. The second pellet was resuspended in 600 µl prewarmed starvation medium, and split into three prewarmed tubes, where cells were incubated in a 41°C Eppendorf incubator for different time points, before immediately being placed on ice to stop endocytosis. Next, starvation medium containing 1:500 Alexa Fluor 647–conjugated donkey anti-mouse antibody was added to cell suspensions. Incubation with secondary antibody was achieved for 30 min. Finally, cells were collected by centrifugation and washed once in PBS before immediate fixation in 4% PFA. Fixation was conducted for 5 min on ice, followed by 15 min at room temperature. Fixed cells were washed twice in PBS and analyzed by BD LSRFortessa analyzer. Data were analyzed using BD FACSDiva or FLowJo software, gating on single intact cells using forward and side scatter.

## Generation of Texas Red-X–conjugated mouse Fab anti-chicken IgM

Mouse anti-chicken IgM antibody was digested into Fab and Fc fragments using Pierce Fab Preparation Kit according to the manufacturer's instructions. The buffer of resulting antibody fragments was exchanged using desalting columns preequilibrated with PBS. Fc fragments were then removed by incubation with protein A agarose beads for 10 min at room temperature.

Successful digestion was verified by SDS-PAGE followed by InstantBlue Coomassie Protein staining. Fluorescent labeling of the resulting Fab fragments was achieved using Texas Red-X Protein Labeling Kit following the manufacturer's instructions. Resulting antibody Fab fragments were concentrated using a centrifugal protein concentrator, before addition of BSA as a stabilizer.

## Live-cell TIRF microscopy

$2 \times 10^6$ cells in logarithmic growth were harvested, washed once in PBS, and incubated on ice for 30 min in 100 µl PBS containing 0.8 µg Texas Red–conjugated mouse Fab anti-chicken IgM. Cells were mixed by pipetting every 10 min. Cells were then pelleted, washed once in ice-cold PBS, and resuspended in 400 µl ice-cold starvation medium. Cells were kept on ice until imaging. To activate cells, 100 µl of cell suspension containing $0.5 \times 10^6$ cells was diluted in 400 µl of prewarmed starvation medium before loading immediately into a well of an Ibidi 4-well glass bottom slide precoated with 10 µg/ml goat polyclonal anti-chicken IgM. Cells were brought into focus as they landed on the glass coverslip and imaged in TIRF mode immediately. Images were collected on a Leica Infinity TIRF microscope (controlled by Leicaa LAS X software) using a 100×/1.47 HC PL Apo Corr TIRF oil objective with 488-nm solid state and 561-nm DPSS TIRF lasers (with 30% laser power and a penetration depth of 100 nm) and an ORCA Flash V4 CMOS camera (Hamamatsu) with 400-ms exposure time. Live imaging was performed at 39.5–41°C.

## Fixed-cell TIRF microscopy

$10 \times 10^6$ cells were collected and washed once in PBS. Cells were incubated in starvation medium for 2 h at a concentration of $2 \times 10^6$ cells/ml. 2 million cells were then collected and labeled with a total of 0.8 µg Texas Red–conjugated mouse Fab anti-chicken IgM in 100 µl PBS on ice for 30 min. Labeled cells were collected and resuspended in 500 µl prewarmed starvation medium and plated onto 35-mm Ibidi dishes precoated with 10 µg/ml of goat polyclonal anti-chicken IgM antibody for 15 min at 41°C, before immediate fixation in 4% PFA at room temperature for 15 min. Membrane permeabilization was achieved by incubation with 0.2% Triton X-100 in PBS for 10 min at room temperature. To visualize actin, cells were stained with phalloidin conjugate for 30 min. Cells were stored in PBS until imaging. Images were acquired at room temperature using a Leica Infinity TIRF microscope set as above.

## Stimulation of BCR in DT40 cells for cellular signaling experiments

$40 \times 10^6$ cells were harvested, washed twice in PBS, and resuspended in starvation medium (RPMI-1640, 1% BSA, 2 mM glutamine, and 50 µM 2-mercaptoethanol). Cells were incubated for 2 h. To stimulate BCR in solution, cells were pelleted and resuspended in starvation medium containing mouse monoclonal anti-chicken IgM antibody at a final concentration of 1 µg/ml. In the case of stimulation with immobilized antibody, 4 million cells in a volume of 1 ml starvation medium were added to an anti-IgM–coated glass-bottom 35-mm Ibidi dish. Cells were lysed at different time points (see Results).

## Measurement of intracellular calcium

Intracellular calcium levels were measured by ratiometric imaging using Fura-2 as in Rahman and Rahman (2017). All $Ca^{2+}$ imaging experiments were performed at room temperature. Ratio fluorescence images were collected on a Nikon Eclipse Ti-S Microscope using a QIClick digital CCD camera (QImaging). Consecutive excitation was provided by a Dual OptoLED Power Supply (Cairn), alternating between both 355-nm ($F_{355}$) and 380-nm ($F_{380}$) LED wavelengths. Images were acquired using MetaFluor (Molecular Devices) Fluorescence Ratio Imaging Software every 5 s. The fluorescence at each time point was extracted for both 355- and 380-nm wavelengths and corrected for autofluorescence. The 355-/380-nm ratios ($F_{355}/F_{380}$) were then calculated to represent intracellular $Ca^{2+}$ levels.

## MS quantification of phosphoinositides

MS quantification was performed on $2.5 \times 10^6$ cells as in Bulley et al. (2016) using a QTRAP 4000 (AB Sciex) mass spectrometer and using the lipid extraction and derivatization method described for cultured cells in Clark et al. (2011), with the modification that C17:0/C16:0 $PI(3,4,5)P_3/PI(4,5)P_2$ (10 ng/0.175 ng) and C17:0/C16:0 PtdIns (100 ng) internal standards were added to primary extracts. Data are shown as response ratios, calculated by normalizing the multiple reaction monitoring–targeted lipid integrated response area to that of a known amount of relevant internal standard. $PIP_3$ response ratios were normalized to the phosphoinositide response ratio to account for any cell input variability, and measurements were conducted in triplicate per experiment.

## Active small GTPases pull-down

Cells were lysed in a buffer containing 25 mM Hepes, pH 7.5, 150 mM NaCl, 1% NP-40, 10 mM $MgCl_2$, 1 mM EDTA, 2% glycerol, and protease inhibitor cocktail. Lysates were cleared by centrifugation at 16,000 $g$ for 30 min at 4°C. The resulting supernatants (containing ~2 mg of total protein) were incubated for 30 min at 4°C with recombinant effector proteins immobilized to glutathione agarose beads. 50 µg of Raf1-RBD, 10 µg of PAK1-PBD, or 10 µg Ral1GDS-RBD was used to enrich active Ras, CDC42, or Rap1, respectively. Bound proteins were then eluted in 2× SDS-PAGE loading buffer and analyzed by immunoblotting.

## Detection of apoptosis by annexin V-FITC staining

Annexin V-FITC staining was performed with or without anti-IgM stimulation for 6 h using FITC Annexin V Apoptosis Detection Kit (BioLegend) according to the manufacturer's instructions. Propidium iodide staining was incorporated in the protocol to assess cell viability. Apoptotic response is expressed as a percentage of cells showing annexin V staining, and data were pooled for early and late apoptosis.

## Image analysis

Fluorescence Images were processed and analyzed using Fiji software (https://fiji.sc). BCR capping (induced by soluble anti-IgM) was quantified using a binary readout, whereby cells were either generated or failed to generate a BCR cap. A BCR cap was defined by at least a twofold increase in the TFI of BCR membrane staining at one side of the cell (opposite to the other side) at ~3 min after stimulation. The BCR capping response was expressed as a percentage of cells scored successful for BCR capping. BCR microclusters were identified using the Laplacian of Gaussians detector in the Fiji plugin TrackMate (Tinevez et al., 2017), with a threshold set to a minimum of 5. To analyze $PI(4,5)P_2$ and F-actin clearance, three regions of enrichment (foci) per cell were identified by the user. The MFI of biosensors within the selected regions were quantified at 1-min intervals starting from ~2 min after cells interacted with the antibody-coated coverslips. MFI quantifications were normalized to the MFI at 2 min. The cell spread area was quantified by using the outer face of the peripheral actin and/or $PI(4,5)P_2$ staining to define the cell edge.

## Statistical analysis

Statistical analyses were conducted using GraphPad Prism 9 Software, and tests used were indicated for each experiment in the corresponding figure legend. Data distribution was assumed to be normal but was not formally tested.

## Online supplemental material

Fig. S1 shows the strategy used to generate INPP5B[Degron/Degron] cells. Fig. S2 shows that OCRL is dispensable for BCR clustering and signaling. Fig. S3 shows $PIP_2$ quantification in INPP5B[Degron/Degron] cells by MS as well as data from an INPP5B[Degron/Degron]-derived cell line stably expressing $PI(4,5)P_2$ biosensor Tubby-EGFP showing that expression of the biosensor does not perturb BCR clustering and signaling. Fig. S4 shows time-lapse TIRF microscopy images of F-actin and BCR in INPP5B[Degron/Degron] cells. In addition, it contains data showing Rap1 effector pulldown assay in INPP5B[Degron/Degron] cells. Fig. S5 shows data suggesting that the defect in BCR capping is dependent on actin dynamics. Videos 1 and 2 correspond to Fig. 1 and show impaired BCR capping and clustering, respectively, in INPP5B-depleted cells. Videos 3 and 4 correspond to Fig. 4 and show $PI(4,5)P_2$ (EGFP-Tubby) dynamics in control and INPP5B-depleted cells, respectively. Videos 5 and 6 correspond to Fig. 6 and show F-actin (LifeAct-EGFP) dynamics in control and INPP5B-depleted cells, respectively. Table S1 lists the source and catalogue numbers for the reagents and antibodies used in this study. Table S2 lists the sequences for the primers used in this study.

## Acknowledgments

We are grateful to Peter March at the Faculty of Biology, Medicine and Health (FBMH) bioimaging facility and to Gareth Howell and David Chapmen at the FBMH flow cytometry facility for their help and advice. We thank Robin F. Irvine (retired, University of Cambridge, Cambridge, UK) for the generous gifts of plasmids, and the electroporation equipment. We also thank Gloria Lopez-Castejon and Rodrigo Diaz Pino (University of Manchester, Manchester, UK) for kindly providing PBMCs.

A. Droubi was supported by a BBSRC research grant (BB/S014799/1) awarded to M. Lowe. C. Wallis and A.D. Sa were supported by a Wellcome Trust PhD studentship (222757/Z/21/Z)

and a Manchester-Singapore A*Star PhD studentship, respectively. K.E. Anderson, L.R. Stephens, and P.T. Hawkins were supported by a BBSRC research grant (BB/PO13384/1). S. Rahman was funded by a PhD studentship from Cambridge Trust.

The authors declare no competing financial interests.

Author contributions: A. Droubi: Conceptualization, methodology, investigation, validation, formal analysis, data curation, visualization, writing (original draft), writing (review & editing). C. Wallis: Methodology, investigation, formal analysis. K.E. Anderson, S. Rahman: Investigation, formal analysis. A.D. Sa: Methodology. T. Rahman, L.R. Stephens, P.T. Hawkins: Resources. M. Lowe: Conceptualization, funding acquisition, supervision, writing (original draft), writing (review & editing).

Submitted: 3 December 2021

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

# Supplemental material

::: JCB
:::

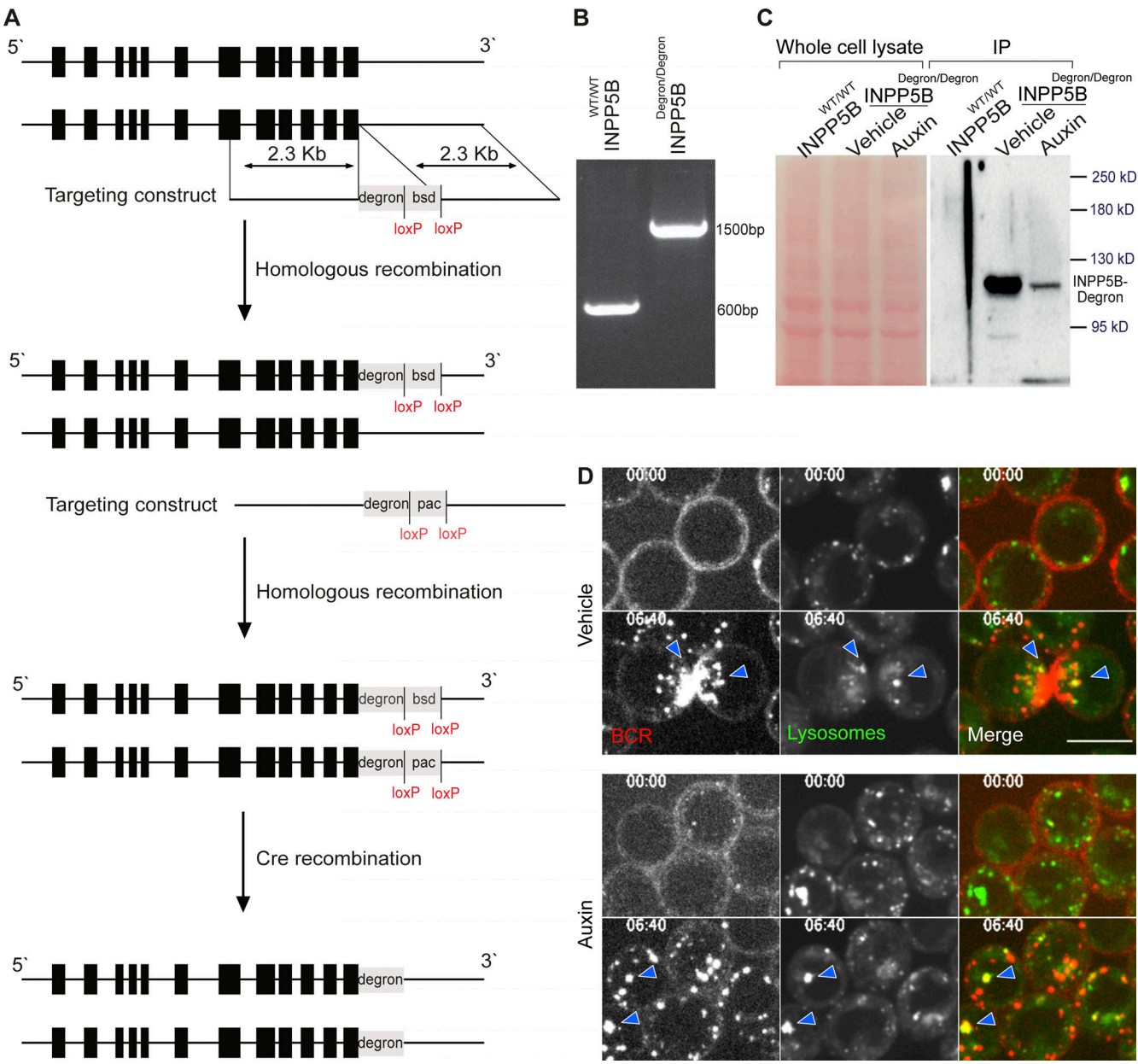

Figure S1. **Generation of INPP5B^Degron/Degron^ cells. (A)** Schematic representation of the chicken *inpp5b* genomic locus and the strategy used to knock in an auxin degron tag at the C-terminus. Exons are shown in black boxes. Two sequential rounds of transfections with targeting constructs bearing different antibiotic resistance genes were performed. bsd, Blasticidin; pac, Puromycin. The antibiotic selection cassettes were removed from the genome by Cre recombination. **(B)** RT-PCR analysis of WT DT40 cells and INPP5B^Degron/Degron^ cells using gene-specific primers flanking the *inpp5b* stop codon. **(C)** Cell-free extracts from WT DT40 cells and INPP5B^Degron/Degron^ cells that were treated with or without auxin for 2 h were subjected to IP using anti-FLAG antibody. INPP5B was detected by WB using anti-poly His. **(D)** Representative confocal images at the indicated time points from INPP5B^Degron/Degron^ cells stimulated with anti-IgM antibody in the presence of LysoTracker Green to visualize lysosomes. Colocalization events are denoted by blue arrowheads. Source data are available for this figure: SourceData FS1.

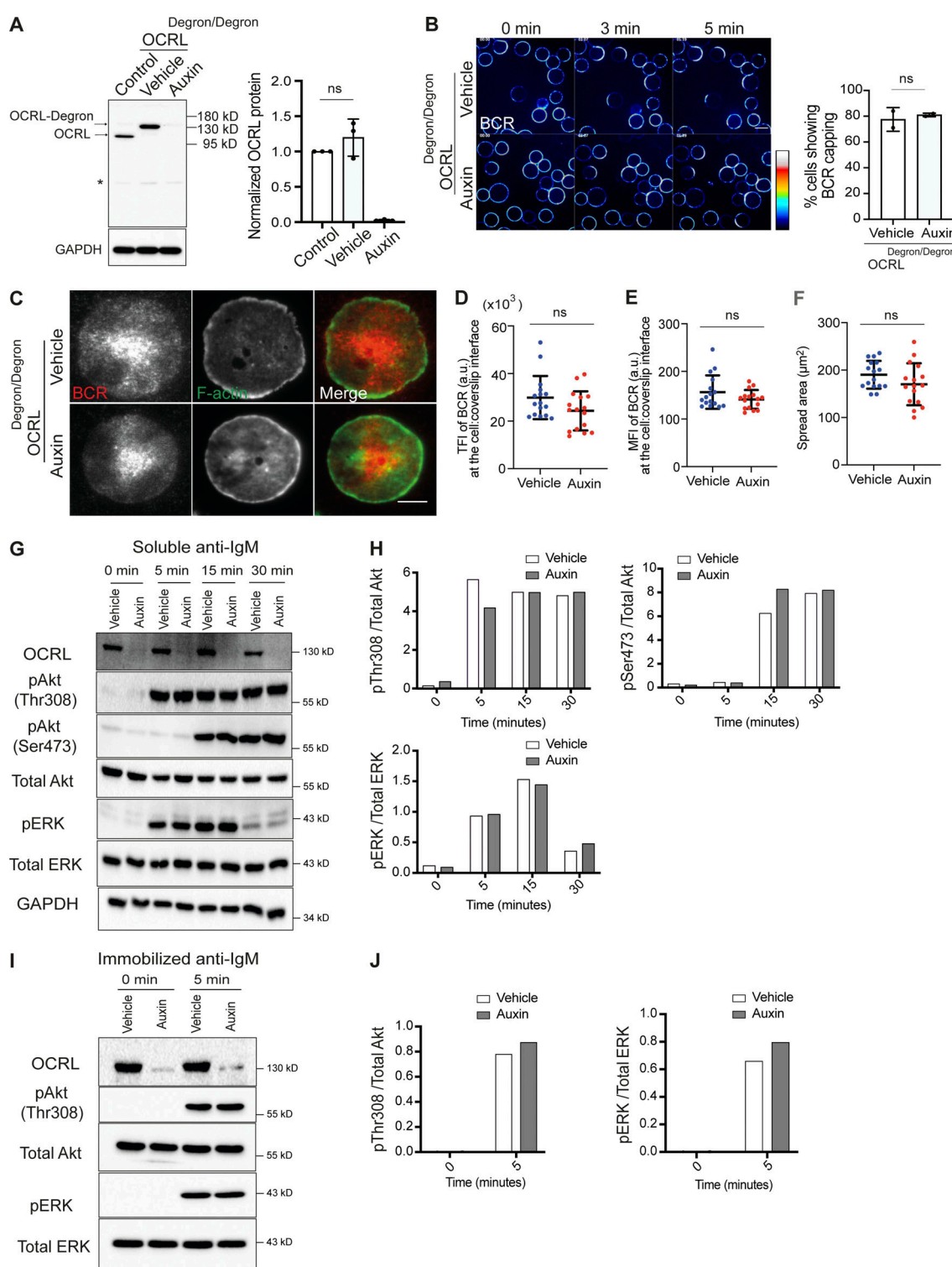

Figure S2. **The inositol 5-phosphatase OCRL is not required for BCR clustering or signaling. (A)** Cell-free extracts from WT DT40 cells and OCRL[-]Degron/Degron cells that were treated with or without auxin for 2 h were blotted with an anti-OCRL antibody. Analysis of three such blots are shown on the right. **(B)** Representative confocal images at the indicated time points from OCRL[Degron/Degron] cells stimulated with Texas Red–conjugated anti-IgM antibody. Analysis of BCR capping is shown on the right. Scale bar, 10 μm. **(C)** Representative TIRF microscopy images from BCR-labeled OCRL[Degron/Degron] cells that were settled on coverslips presenting surrogate antigens for 15 min. F-actin was stained with Phalloidin. Scale bar, 5 μm. **(D–F)** Statistical analyses of TFI (BCR), spread area, and MFI (BCR) from vehicle-treated and auxin-treated OCRL[Degron/Degron] cells in D, E, and F, respectively. The data represent mean ± SD of 15 cells. **(G)** Protein extracts from OCRL[Degron/Degron] cells stimulated in solution were blotted for Akt and ERK phosphorylation at the indicated time points. **(H)** Quantification of the blots (from a single experiment) by densitometry. **(I)** Protein extracts from OCRL[Degron/Degron] cells stimulated on glass coverslips were blotted for Akt and ERK phosphorylation. **(J)** Quantification of the blots (from a single experiment) by densitometry. Source data are available for this figure: SourceData FS2.

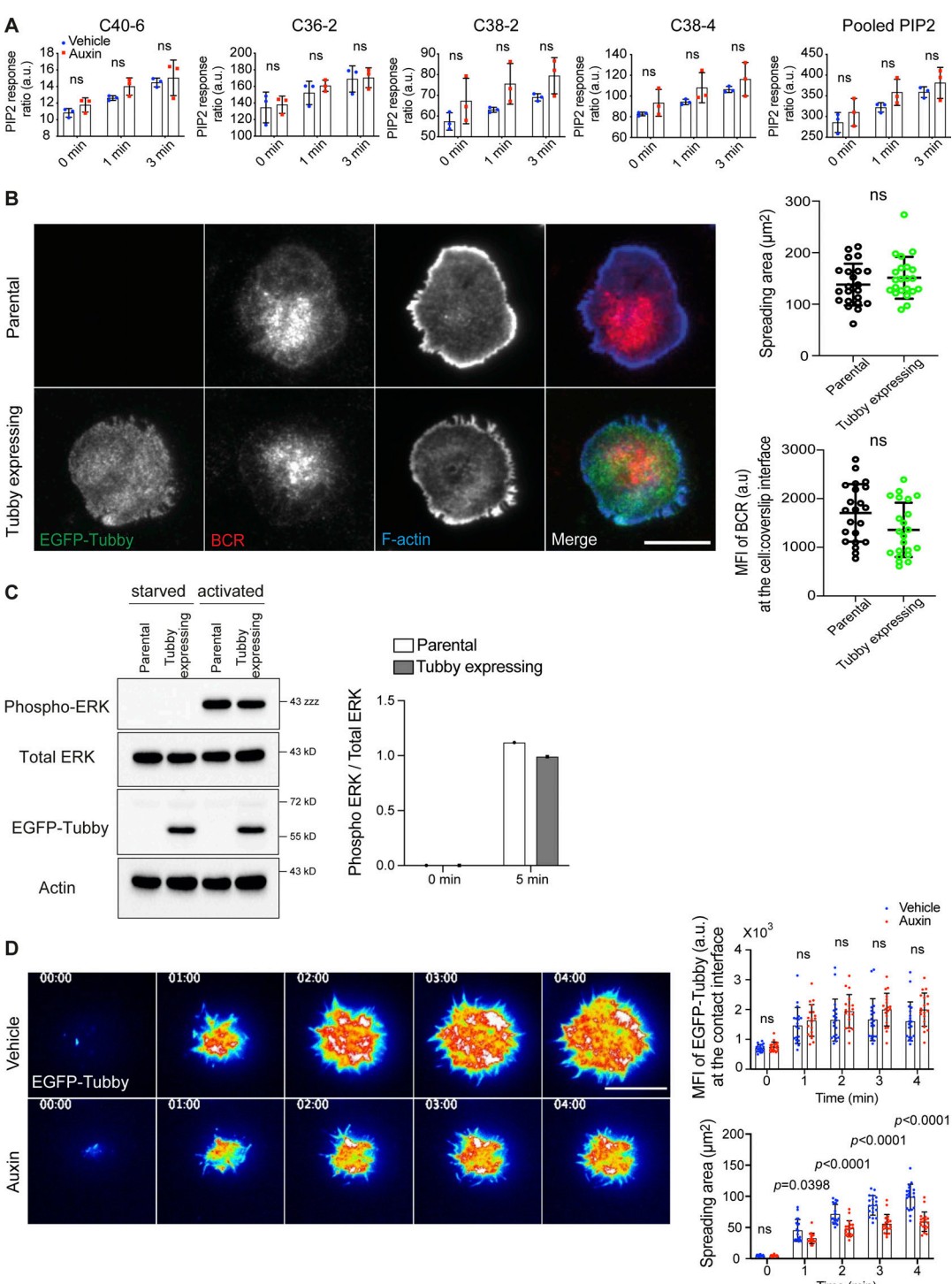

Figure S3. **The expression of EGFP-Tubby does not perturb BCR clustering or signaling. (A)** PIP$_2$ levels in INPP5B-depleted cells. INPP5B[Degron/Degron] cells were stimulated in solution for the times indicated, before analysis by MS. PIP$_2$ levels are expressed relative to PI, and data for the C40:6, C36:2, C38:2, and C38:4 species or a pool of all four are shown separately. Data from three independent experiments were analyzed by two-way ANOVA. **(B)** Representative TIRF microscopy images from parental INPP5B[Degron/Degron] cells or a derivative clone expressing EGFP-Tubby that were settled on coverslips presenting surrogate antigens for 15 min. F-actin was stained with Alexa Fluor 488–Phalloidin. Scale bar, 10 μm. Statistical analyses of the spread area and the MFI of BCR staining at the contact interface are shown on the right. The data represent mean ± SD of 22 cells. **(C)** Western blot analysis of phospho-ERK in lysates from INPP5B[Degron/Degron] cells stably expressing EGFP-Tubby. The parental cells were used as a control. The quantification of blots (from a single experiment) is shown on the right. **(D)** Representative TIRF microscopy images at the indicated time points of INPP5B[Degron/Degron] cells expressing a fluorescent biosensor (tubby) after stimulation on antibody-coated coverslips. Scale bar, 10 μm. Images are pseudo-colored. See Videos 3 and 4 for the complete time-lapse TIRF microscopy images. Analysis of the intensity of PI(4,5)P$_2$ per area (MFI; top) and the spreading area (bottom) at the indicated time points in vehicle- vs. auxin-treated cells (n = 20) are shown on the right. Data was analyzed by two-way ANOVA, and P values were calculated using Sidak multiple comparisons test. Bars represent mean ± SD. Source data are available for this figure: SourceData FS3.

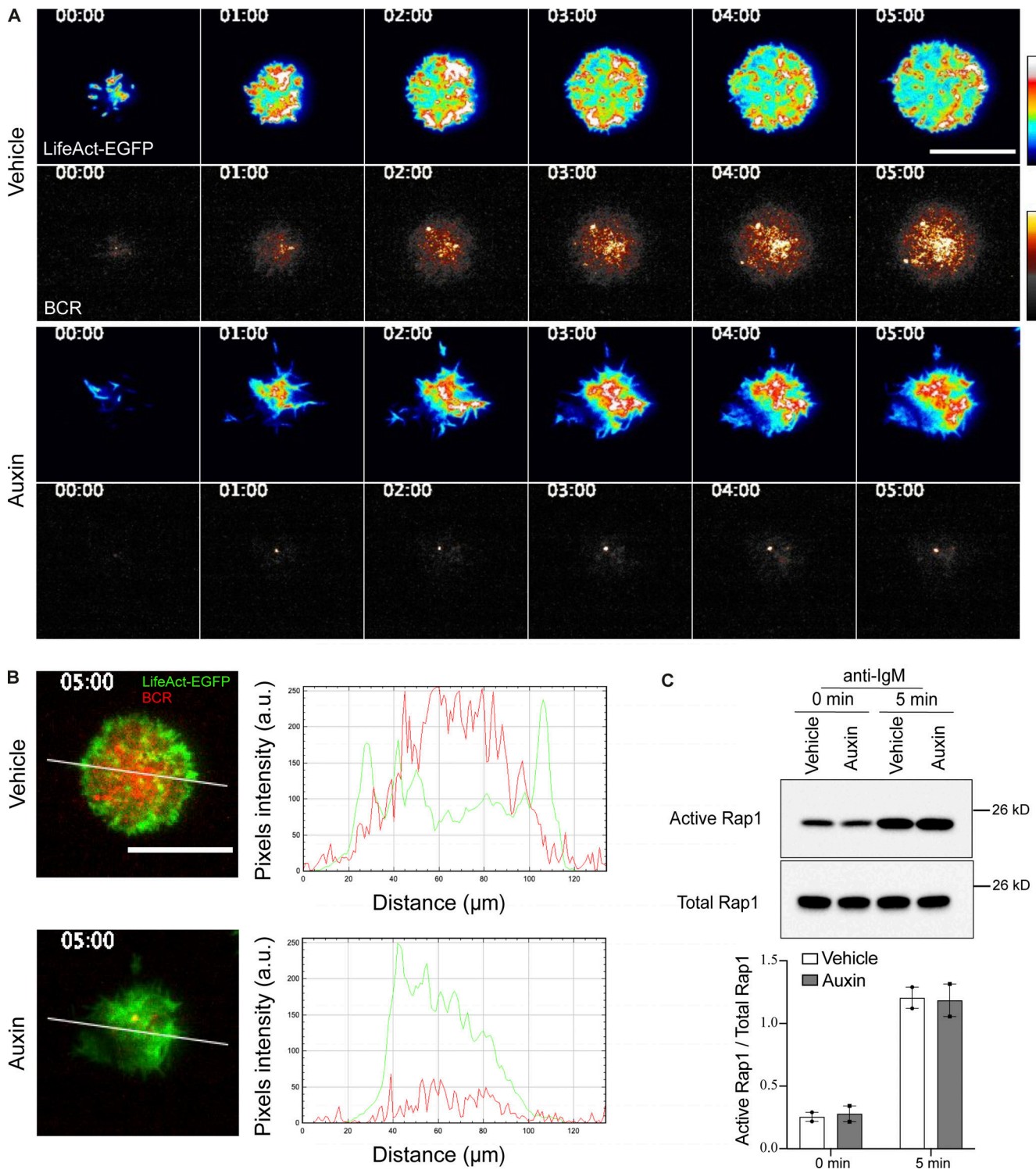

Figure S4.   **F-actin turnover at the cell-coverslip contact interface is required for BCR clustering. (A)** Two-color time-lapse TIRF microscopy images of F-actin (LifeAct-EGFP) and BCR in vehicle-treated (top rows) and auxin-treated (bottom rows) INPP5B$^{Degron/Degron}$ cells that were stimulated on antibody-coated glass. Images are pseudo-colored. **(B)** Representative TIRF microscopy images showing BCR and F-actin (LifeAct-EGFP) of vehicle- vs. auxin-treated INPP5B$^{Degron/Degron}$ cells at 5 min. Fluorescence intensity profiles of F-actin and BCR on the white line in the TIRF microscopy images are shown on the right. **(C)** GTP-bound Rap1 was detected using RalGDS-RBD GST pulldown followed by blotting against Rap1. Quantification of two such experiments by densitometry is shown below. GTP levels of Rap1 are expressed relative to total Rap1. Source data are available for this figure: SourceData FS4.

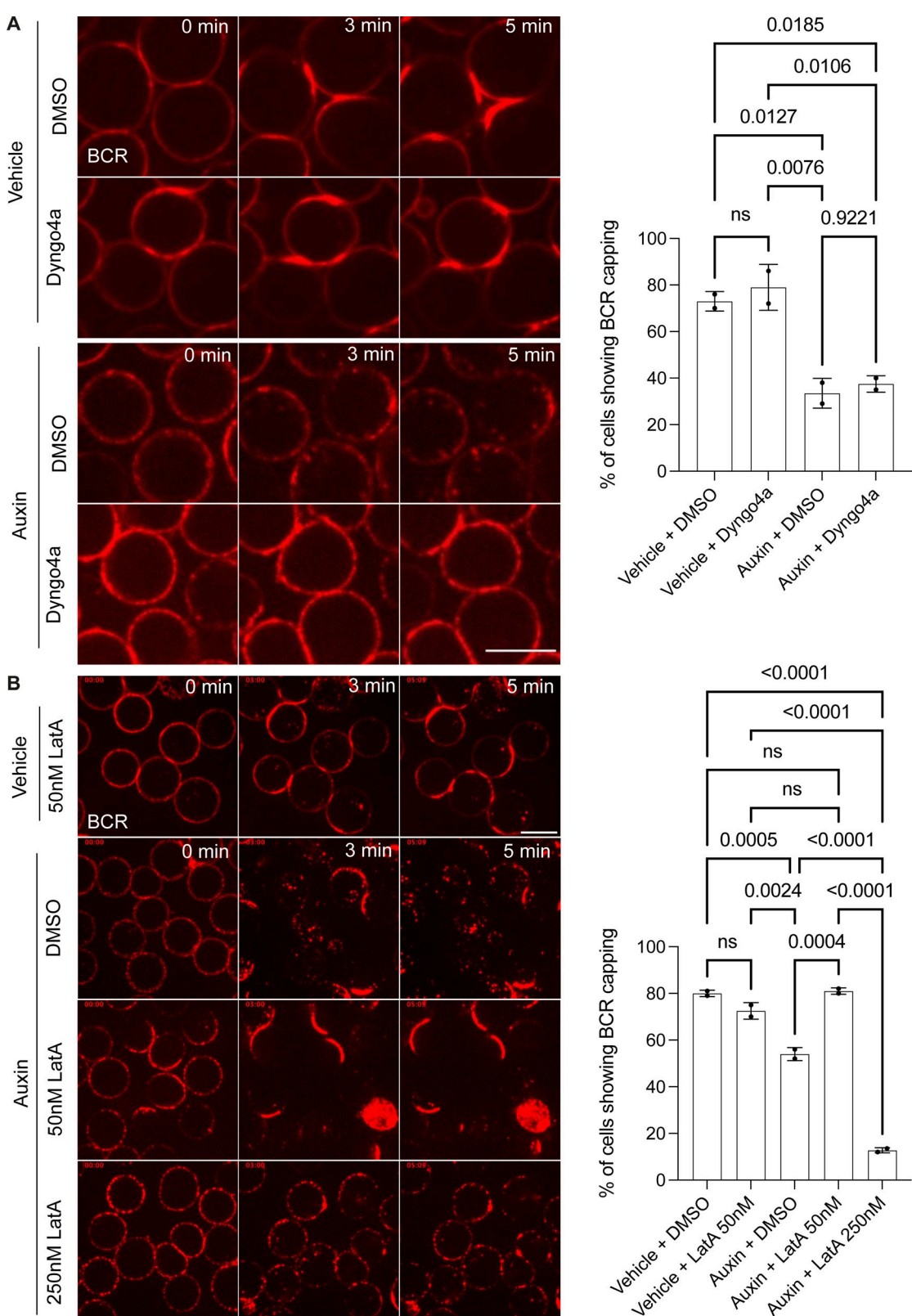

Figure S5. **The defect in BCR capping in INPP5B-depleted cells is actin dependent. (A)** Representative confocal images at the indicated time points from control and INPP5B-depleted cells stimulated with Texas Red–conjugated anti-IgM antibody in the presence of DMSO vs. Dyngo-4a (50 μM). Scale bar, 10 μm. Quantification of BCR capping at ∼3 min is shown on the right. Data from two independent experiments were analyzed by one-way ANOVA, and P values were calculated using Tukey's multiple comparisons test. Error bars indicate SD. **(B)** Representative confocal images at the indicated time points from INPP5B-depleted cells stimulated with Texas Red–conjugated anti-IgM antibody in the presence of the indicated concentrations of LatA or DMSO. Scale bar, 10 μm. Quantification of BCR capping at ∼3 min is shown on the right. Statistical analysis was performed as above on data pooled from two independent experiments.

Video 1.   **BCR capping in INPP5B<sup>Degron/Degron</sup> cells.** Related to Fig. 1 B. Vehicle- or auxin-treated INPP5B<sup>Degron/Degron</sup> cells stimulated with Texas Red–conjugated anti-IgM antibody were imaged using time-lapse spinning disc confocal microscopy. The video is displayed at 20 frames/s. Scale bar, 10 µm.

Video 2.   **BCR clustering in INPP5B<sup>Degron/Degron</sup> cells stimulated on glass coverslips presenting surrogate antigen.** Related to Fig. 1. Vehicle- or auxin-treated INPP5B<sup>Degron/Degron</sup> cells, prelabeled for BCR, were imaged by TIRF microscopy during their interaction with antibody-coated coverslips. The video is displayed at 100 frames/s. Scale bar, 10 µm.

Video 3.   **PI(4,5)P$_2$ (EGFP-Tubby) dynamics in INPP5B<sup>Degron/Degron</sup> control cells.** Related to Fig. 4. Vehicle-treated INPP5B<sup>Degron/Degron</sup> cells, expressing EGFP-Tubby, were imaged by TIRF microscopy during their interaction with antibody-coated coverslips. The video is displayed at 20 frames/s. Scale bar, 5 µm.

Video 4.   **PI(4,5)P$_2$ (EGFP-Tubby) dynamics in INPP5B-depleted cells.** Related to Fig. 4. Auxin-treated INPP5B<sup>Degron/Degron</sup> cells, expressing EGFP-Tubby, were imaged by TIRF microscopy during their interaction with antibody-coated coverslips. The video is displayed at 20 frames/s. Scale bar, 5 µm.

Video 5.   **F-actin (LifeAct-EGFP) dynamics in INPP5B<sup>Degron/Degron</sup> control cells.** Related to Fig. 6. Vehicle-treated INPP5B<sup>Degron/Degron</sup> cells, expressing LifeAct-EGFP, were imaged by TIRF microscopy during their interaction with antibody-coated coverslips. The video is displayed at 20 frames/s. Scale bar, 10 µm.

Video 6.   **F-actin (LifeAct-EGFP) dynamics in INPP5B-depleted cells**. Related to Fig. 6. Auxin-treated INPP5B<sup>Degron/Degron</sup> cells, expressing LifeAct-EGFP, were imaged by TIRF microscopy during their interaction with antibody-coated coverslips. The video is displayed at 20 frames/s. Scale bar, 10 µm.

**Provided online are Table S1 and Table S2. Table S1 lists the source and catalogue numbers for the reagents and antibodies used in this study. Table S2 lists the sequences for the primers used in this study.**

