## [Peer Review File · The Journal of Cell Biology]

The inositol 5-phosphatase INPP5B regulates B cell receptor clustering and signaling

Alaa Droubi, Connor Wallis, Karen Anderson, Saifur Rahman, Aloka de Sa, Taufiq Rahman, Len Stephens, Phillip Hawkins, and Martin Lowe

Corresponding Author(s): Martin Lowe, University of Manchester

Review Timeline:

Submission Date:	2021-12-03
Editorial Decision:	2022-01-10
Revision Received:	2022-05-27
Editorial Decision:	2022-06-27
Revision Received:	2022-07-01

Monitoring Editor: Ana-María Lennon-Dumenil

Scientific Editor: Andrea Marat

Transaction Report:

DOI: <https://doi.org/10.1083/jcb.202112018>

January 10, 2022

Re: JCB manuscript #202112018

Prof. Martin Lowe
University of Manchester
Faculty of Biology, Medicine and Health
Michael Smith Building
Oxford Road
Manchester M13 9PT
United Kingdom

Dear Prof. Lowe,

Thank you for submitting your manuscript entitled "The inositol 5-phosphatase INPP5B regulates clustering and activation of the B Cell Receptor". The manuscript was assessed by expert reviewers, whose comments are appended to this letter. We invite you to submit a revision if you can address the reviewers' key concerns, as outlined here.

As you will see, the reviewers all express enthusiasm regarding the identification of INPP5B as being involved in B cell activation and synapse assembly. However, they have provided constructive feedback to further improve your study. In particular, we agree that validation in another B cell type such as primary B cells is essential, as are all of the other relatively minor points of reviewers #2 and #3. Reviewer #1 has also provided suggestions to extend the mechanistic insight to further examine the effects of INPP5B depletion on BCR signaling and to use mobile antigen to provide a more physiologically relevant examination. We agree that all of reviewer #1's suggestions are interesting and would welcome data addressing all of these issues. In particular it should be feasible to link INPP5B to a longer term readout of B cell activation. However, fully addressing all of these comments may be outside the scope of a typical JCB paper and not feasible within our standard 3-4 month revision period. Therefore, once you have had time to consider their comments we encourage you to prepare a revision plan outlining what experiments you are able to conduct, so that we may provide feedback on suitability for resubmission to JCB. Please let us know if you have any questions or would like to discuss further.

GENERAL GUIDELINES:

Text limits: Character count for an Article is < 40,000, not including spaces. Count includes title page, abstract, introduction, results, discussion, acknowledgments, and figure legends. Count does not include materials and methods, references, tables, or supplemental legends.

Figures: Articles may have up to 10 main text figures. Figures must be prepared according to the policies outlined in our Instructions to Authors, under Data Presentation, <https://jcb.rupress.org/site/misc/ifora.xhtml>. All figures in accepted manuscripts will be screened prior to publication.

Supplemental information: There are strict limits on the allowable amount of supplemental data. Articles may have up to 5 supplemental figures. Up to 10 supplemental videos or flash animations are allowed. A summary of all supplemental material should appear at the end of the Materials and methods section.

Please note that JCB now requires authors to submit Source Data used to generate figures containing gels and Western blots with all revised manuscripts. This Source Data consists of fully uncropped and unprocessed images for each gel/blot displayed in the main and supplemental figures. Since your paper includes cropped gel and/or blot images, please be sure to provide one Source Data file for each figure that contains gels and/or blots along with your revised manuscript files. File names for Source Data figures should be alphanumeric without any spaces or special characters (i.e., SourceDataF#, where F# refers to the associated main figure number or SourceDataFS# for those associated with Supplementary figures). The lanes of the gels/blots should be labeled as they are in the associated figure, the place where cropping was applied should be marked (with a box), and molecular weight/size standards should be labeled wherever possible.

As you may know, the typical timeframe for revisions is three to four months. However, we at JCB realize that the implementation of social distancing measures that limit spread of COVID-19 also pose challenges to scientific researchers. Therefore, JCB has waived the revision time limit. Please note that papers are generally considered through only one revision cycle, so any revised manuscript will likely be either accepted or rejected.

Thank you for this interesting contribution to Journal of Cell Biology. You can contact us at the journal office with any questions, cellbio@rockefeller.edu or call (212) 327-8588.

Sincerely,

Ana-María Lennon-Dumenil, PhD
Monitoring Editor

Andrea L. Marat, PhD
Senior Scientific Editor

Journal of Cell Biology

Reviewer #1 (Comments to the Authors (Required)):

Summary: The INPP5B phosphatase dephosphorylates the 5 position of PI(4,5)P2, as well as PIP3 and IP3. This manuscript presents novel data showing that the enzymatic activity of INPP5B promotes BCR clustering, early BCR signaling responses, and the BCR-induced actin remodeling that drives B cell spreading while limiting the endocytosis of clustered BCRs. These responses are all regulated by a network of enzymes that act on PI(4,5)P2 and PIP3 (e.g. PLCg2, PI3K, SHIP, PTEN) but this is the first report that INPP5B shapes the inter-related processes of phosphoinositide flux, BCR microcluster formation and signaling, and actin dynamics. The manuscript convincingly shows that INPP5B modulates the changes in PI(4,5)P2 and PIP3 levels caused by BCR engagement and that depleting INPP5B results in impaired actin dynamics and reorganization during BCR-driven spreading. A particularly interesting finding is that INPP5B controls PI(4,5)P2 localization in B cells spreading on immobilized anti-Ig and that INPP5B-dependent actin clearance supports the formation of BCR microclusters. Although the mechanisms by which INPP5B impacts actin dynamics in B cells are not fully explored, the authors show that INPP5B modulates the activation states of cofilin and ezrin, proteins that have been linked to initial BCR-induced changes in actin dynamics and reorganization.

Overall comments: The manuscript presents novel and interesting findings. The knock-in of an INPP5B-degron construct is an elegant approach that enables the rapid depletion (1-2 hr) of INPP5B. The manuscript is very well written. The experiments are all very well done, both representative experiments and compiled data from multiple independent experiments are presented, and extensive controls have been performed. The data are clear and appropriate statistical analyses have been used. The microscopy data are excellent. The multi-color real-time TIRF imaging of BCR microclusters, F-actin, and PI(4,5)P2 at the cell-substrate contact site is particularly impressive and provides key insights into the order of events.

The manuscript could be strengthened considerably by some key extensions of the work. Specifically, the effects of INPP5B depletion on BCR signaling and actin regulatory pathways are incompletely explored. It is not clear whether INPP5B selectively modulates the activation of a specific subset of BCR signaling targets or has more global effects. A more complete analysis may be very revealing.

A key limitation of the manuscript is that all of the experiments were done with a single B cell line, the chicken DT-40 B cell line. Confirming some of the key findings in primary mouse B cells or a mammalian B cell line would better allow one to generalize the findings to all B cells.

The manuscript would also be strengthened by linking INPP5B to a longer-term readout of B cell activation (e.g. transcription factor activation or BCR-induced apoptosis).

Finally, the experiments employing immobilized anti-Ig are very informative but the physiological relevance would be greatly increased by showing that INPP5B depletion alters B cell spreading and BCR microcluster formation in response to membrane-bound anti-Ig. A situation where the surrogate antigen is mobile would better reflect the process of B cell activation by an antigen-presenting cell. If the authors are in a position to do such experiments, it would be an important extension of the work.

Specific suggestions for further experimentation:

1. Effects of INPP5B depletion on BCR signaling: It is not clear whether INPP5B depletion impairs most BCR signaling responses or only a select subset of downstream responses that are regulated by PI(4,5)P2 and PIP3. This is further complicated by positive and negative feedback loops and by actin-dependent BCR clustering, which amplifies BCR signaling. Hence, a more comprehensive analysis of BCR signaling is warranted.
 - a) To assess proximal/initial BCR signaling, CD79 phosphorylation (and perhaps also Syk phosphorylation) should be assessed. The commercial pCD79a antibody may work for chicken B cells. Alternatively, if the cells are solubilized with digitonin, mIg-CD79a/b complexes are preserved and could be immunoprecipitated via the stimulating anti-Ig antibody, and then probed with anti-pTyr.
 - b) Immunoblotting total cell lysates with anti-pTyr would provide an overall sense of whether BCR signaling responses are globally or selectively reduced in INPP5B-depleted cells. Because, altering phosphoinositide levels could affect feedback loops that amplify or limit BCR signaling, a dose-response for the anti-Ig stimulating antibody (just the 3-minute time point would be sufficient) would reveal whether INPP5B depletion alters the sensitivity of BCR signaling.
 - c) Figure 3F shows that fewer cells undergo Ca²⁺ oscillations when INPP5B is depleted and that the oscillations occur at a reduced rate. Is PLC β 2-mediated PI(4,5)P2 hydrolysis impaired in the INPP5B-depleted cells? Hence, it would be of interest to measure levels of IP3 and DAG, especially since INPP5B may also act on IP3. On page 12, in the discussion, the authors do mention that loss of INPP5B prevents PI(4,5)P2 hydrolysis in other systems.
2. Effects of INPP5B depletion on key actin regulators that are targets of BCR signaling: In addition to cofilin and ezrin, the altered phosphoinositide levels and impaired BCR signaling in INPP5B-depleted cells likely affects multiple nodes in the actin regulatory network. BCR engagement stimulates activation of the Rac2, Cdc42, and Rap1 GTPases, all of which are essential for BCR-induced actin turnover and remodeling. It would be good to assess the activation of these central regulators of actin dynamics using effector pull-down assays, as done for Ras.
3. Confirming several of the key findings in primary mouse B cells or mouse/human B cell lines is important. LPS-stimulated primary mouse B cells can be readily transfected with siRNA and spread well on immobilized anti-Ig. Alternatively, a mouse or human B cell line could be transfected with control versus INPP5B siRNA.
4. It would be good to show that the reduced BCR early signaling in INPP5B-depleted cells have functional consequences and impact longer-term responses that relate to B cell activation or selection.
 - a) If phospho-specific antibodies that recognize the chicken proteins exist, it should be possible to assess the phosphorylation of NF- κ B, NF-AT, or FOXO1 in control versus INPP5B-depleted DT-40 cells after BCR engagement. If such reagents do not exist, this could be investigated in mouse or human B cell lines transfected with control versus INPP5B siRNA.
 - b) DT40 cells have been reported to undergo apoptosis after treatment with anti-Ig antibodies and this response is modulated by the level of anti-apoptotic Akt activation (Pogue et al. J Immunol 165:1300, 2000). Would the decreased Akt T308 phosphorylation in INPP5B-depleted DT40 cells make them more susceptible to anti-Ig-induced apoptosis, e.g. at lower concentrations of anti-Ig? Or would impaired BCR signaling and increased BCR endocytosis (leading perhaps to less prolonged BCR signaling) result in less apoptosis? These could be simple but informative experiments.
5. If it is possible to do so, some initial experiments showing that INPP5B depletion alters immune synapse formation (B cell spreading, BCR microcluster formation, cSMAC formation) in response to membrane-bound anti-Ig would address the physiological relevance of the findings obtained using immobilized anti-Ig.

Minor comments to address in the text:

1. It would be good to replace the term "BCR activation" in the title (and elsewhere) with "BCR signaling". The term "BCR activation" is more often used to describe the dissociation of autoinhibited BCR dimers (as in Reth's Dissociation-Activation Model), followed by phosphorylation of the BCR ITAMs.
2. Page 7. What is the evidence that capping is important for, or amplifies, BCR signaling? Classically, capping was considered to be a precursor to large-scale BCR endocytosis at later time points (e.g. 30 min).
3. Page 7. With regard to Fig. 3A, the authors should state that Akt T308 phosphorylation was significantly reduced at 5 min but not at 15 min or 3 min. This raises the possibility that this response is "delayed", more so than "reduced".

4. Page 7 and page 10. The term "surface antigen" should be avoided. "Immobilized anti-Ig" or "coverslip-bound anti-Ig" would be more appropriate". Reth and others have shown that anti-Ig Abs that bind to the Fc regions of mIg activate BCRs in a fundamentally different manner than ligands that bind to the antigen-binding site in the mIg subunit of the BCR.
5. Is it known whether INPP5B associates constitutively with the inner face of the plasma membrane or whether it is recruited to the plasma membrane by BCR signaling or recruited to nascent BCR clusters? This is a point that could be discussed with regard to the clearance of PI(4,5)P2 that precedes the appearance of larger BCR microclusters.
6. Page 12. The authors mention that loss of INPP5B prevents PI(4,5)P2 hydrolysis in other systems. Have these other publications suggested a mechanism by which INPP5B promotes PLCg-mediated hydrolysis of PI(4,5)P2?
7. Page 13. Is it clear that the reduced levels of PIP3 in INPP5B-depleted cells are due to attenuated PI3K signaling? How would INPP5B, which dephosphorylates PI(4,5)P2 and perhaps also PIP3, contribute to activating PI3K or preserving PIP3 levels via other mechanisms?
8. Page 14. Ref. 56 refers to an INPP5B knockout mouse. It might be worthwhile to refer to this in the introduction and add that immune cell development and function has not been studied in these mice (or if it has, to summarize the findings).

Reviewer #2 (Comments to the Authors (Required)):

"The inositol 5-phosphatase INPP5B regulates clustering and activation of the B cell receptor" focuses on the mechanisms of antigen-driven F-actin remodeling and BCR clustering. Using an elegant auxin-induced degron system, Droubi et al. find that DT40 B cells lacking INPP5B exhibit defects in synaptic F-actin remodeling, antigen capping, and BCR signaling. They go on to provide evidence that the catalytic activity of INPP5B is critical for its function in this context. Finally, they present imaging experiments and biochemical analyses that a) indicate that INPP5B dependent F-actin clearance is required for BCR cluster formation and b) suggest that INPP5B remodels F-actin via cofilin and ezrin, two established regulators of the synaptic cytoskeleton in B cells. This paper is of interest to the readers of JCB because it reveals a previously unappreciated mechanism of B cell activation and synapse assembly, with potentially important implications for the therapeutic control of B cell function. The data are convincing, and the authors have done a good job of explaining and interpreting their results. I have only a few concerns, which the authors should be able to address in a revision.

- 1) The most obvious short-coming of the study is that it relies completely on the DT40 cell line. The authors should confirm key results (such as the INPP5B loss-of-function phenotype) in primary B cells.
- 2) It is perplexing (but also quite interesting) that INPP5B is absolutely required for PIP2 depletion in the early synapse, but that PLCg, which is also recruited to the synapse and hydrolyzes PIP2, is not. Perhaps the two enzymes are differentially localized? This is something the authors could address quite easily using fluorescently labelled constructs or immunocytochemistry.
- 3) Can the anti-correlation between BCR microclusters and PIP2 (Figure 5) be quantitated directly?
- 4) The INPP5B G502D phenotype (Figure 3) is more subtle than the full KO phenotype (Figures 1 and 2). This implies at least some phosphatase independent function for INPP5B, correct?
- 5) The low dose LatA strategy shown in Figure S6B is clever. What happens when wild type cells are treated with the same concentrations of LatA?
- 6) In the Discussion, it is stated that INPP5B-induced F-actin remodeling is "mediated through the activation and inactivation of . . . cofilin and . . . ezrin, respectively". The authors do not actually show this, however. They should either qualify this statement (e.g. "our results are consistent with the model in which. . .") or incorporate some sort of ezrin and/or cofilin perturbation approach to test the hypothesis directly.
- 7) It would be interesting to know how the authors think that INPP5B activity is coupled to antigen recognition by the BCR. They should speculate about this in the discussion. Is INPP5B recruited to Igalpha or Igbeta? Is this recruitment ITAM dependent? Alternatively, might INPP5B lower PIP2 levels constitutively, and could this explain the observed loss-of-function phenotypes?

Reviewer #3 (Comments to the Authors (Required)):

The activation of B cells is a necessary step in the production of antibodies. B cell responses begin with antigen engagement of the B cell receptor (BCR), which triggers BCR clustering, signalling, immune synapse formation, and antigen endocytosis. All steps of this process require actin remodelling. Despite the importance of actin in regulating these key first events in B cell

activation, the mechanisms are not well understood.

This paper identifies a role for the inositol 5 phosphatase INPP5B in each step outlined above. Through a series of well constructed experiments, the authors find that, through hydrolysis of PIP₂, INPP5B promotes cofilin-mediated actin severing and ezrin-mediated disruption of connections between the actin cytoskeleton and plasma membrane, which together promote BCR clustering and B cell spreading on antigen-coated substrates.

The paper addresses an important question in B cell biology. It is very clearly written and the findings in my opinion make a very nice contribution to our understanding of B cell activation. Overall I think the conclusions made in the paper are supported by the results, but there are several cases where I think the data analysis methods need to be more clearly communicated. Specific points are listed below.

Major points / questions

- Control conditions are indicated as 'Vehicle' throughout the text and Figures, but it is not clear to me what the vehicle is. Is it DMSO? In line with this, were control cells treated with auxin at any point to demonstrate that it does not impact cell behaviour in the absence of the auxin degron tag?
- Figures 1B and 3B show the % cells showing BCR capping. How was this quantified? Was it a manual identification of 'yes' or 'no' for each cell, or was a quantitative approach used?
- It would be helpful to have a bit more information about the quantitation in Figure 5. Figures 5C and 5D show the mean fluorescence intensity of PIP₂ and BCR foci, respectively. Are the foci in each figure the same spot, so that the data show the decrease in PIP₂ signal and increase in BCR signal over time in the same small region? Also, Figure 5E shows the PIP₂/BCR intensity ratio. Does the ratio take into account the mean fluorescence intensity throughout the entire synapse, or is it the ratio at the level of the foci?
- Figure 6B shows the analysis of F-actin clearance in B cell synapses. Do the data indicate the percentage of cells in each condition that had cleared actin from the cell centre? If yes, how was this determined? Even in the control cells in Figure 6A, the actin distribution looks relatively homogeneous by 6 minutes, rather than enriched in the periphery and depleted in the middle.

Minor points

- Small grammar correction page 13: "... And hence INPP5B may could also impact..." may and could are redundant
- On page 31, the citation Tinevez is not formatted into the bibliography
- Page 35, Figure 5: spelling error, should be INPP5B (not INPPB)

Editor comments:

As you will see, the reviewers all express enthusiasm regarding the identification of INPP5B as being involved in B cell activation and synapse assembly. However, they have provided constructive feedback to further improve your study. In particular, we agree that validation in another B cell type such as primary B cells is essential, as are all of the other relatively minor points of reviewers #2 and #3.

RESPONSE: To provide validation in another B cell type, we have analyzed the BCR response in two human cell models, primary human B cells isolated from blood cones, and the Ramos human B cell line. We used both chemical inhibition of INPP5B and CRISPR/Cas9-mediated knockout to show that cell spreading in response to BCR stimulation is reduced upon INPP5B loss of function. These new data have been included in a new main figure in the manuscript. We have also addressed all of the more (relatively) minor points raised by reviewers #2 and #3.

Reviewer #1 has also provided suggestions to extend the mechanistic insight to further examine the effects of INPP5B depletion on BCR signaling and to use mobile antigen to provide a more physiologically relevant examination. We agree that all of reviewer #1's suggestions are interesting and would welcome data addressing all of these issues. In particular it should be feasible to link INPP5B to a longer term readout of B cell activation. However, fully addressing all of these comments may be outside the scope of a typical JCB papers and not feasible within our standard 3-4 month revision period.

RESPONSE: We have performed a more thorough investigation of the BCR signaling response in INPP5B-depleted cells. The results indicate a number of changes that point to alterations in both proximal and distal signaling. We have also observed changes in Cdc42 activity that are likely important for the altered actin dynamics we see in the INPP5B-depleted cells. These new data have been included in the revised version of the paper. We have also investigated a longer-term readout of BCR stimulation, namely apoptosis, and find that it is also reduced upon INPP5B depletion. This supports a role for INPP5B in physiological changes downstream of the BCR. This new data is also included in the revision. We have not been able to perform experiments on mobile anti-Ig e.g. using planar bilayers as the technology is not yet established in the lab, and feel that it lies beyond the scope of the current study.

Reviewer comments

Reviewer #1 (Comments to the Authors (Required)):

Summary: The INPP5B phosphatase dephosphorylates the 5 position of PI(4,5)P2, as well as PIP3 and IP3. This manuscript presents novel data showing that the enzymatic activity of INPP5B promotes BCR clustering, early BCR signaling responses, and the BCR-induced actin remodeling that drives B cell spreading while limiting the endocytosis of clustered BCRs. These responses are all regulated by a network of enzymes that act on PI(4,5)P2 and PIP3 (e.g. PLCg2, PI3K, SHIP, PTEN) but this is the first report that INPP5B shapes the inter-related processes of phosphoinositide flux, BCR microcluster formation and signaling, and actin dynamics. The manuscript convincingly shows that INPP5B modulates the changes in PI(4,5)P2 and PIP3 levels caused by BCR engagement and that depleting INPP5B results in impaired actin dynamics and reorganization during BCR-driven spreading. A particularly interesting finding is that INPP5B controls PI(4,5)P2 localization in B cells spreading on

immobilized anti-Ig and that INPP5B-dependent actin clearance supports the formation of BCR microclusters. Although the mechanisms by which INPP5B impacts actin dynamics in B cells are not fully explored, the authors show that INPP5B modulates the activation states of cofilin and ezrin, proteins that have been linked to initial BCR-induced changes in actin dynamics and reorganization.

Overall comments: The manuscript presents novel and interesting findings. The knock-in of an INPP5B-degron construct is an elegant approach that enables the rapid depletion (1-2 hr) of INPP5B. The manuscript is very well written. The experiments are all very well done, both representative experiments and compiled data from multiple independent experiments are presented, and extensive controls have been performed. The data are clear and appropriate statistical analyses have been used. The microscopy data are excellent. The multi-color real-time TIRF imaging of BCR microclusters, F-actin, and PI(4,5)P2 at the cell-substrate contact site is particularly impressive and provides key insights into the order of events.

The manuscript could be strengthened considerably by some key extensions of the work. Specifically, the effects of INPP5B depletion on BCR signaling and actin regulatory pathways are incompletely explored. It is not clear whether INPP5B selectively modulates the activation of a specific subset of BCR signaling targets or has more global effects. A more complete analysis may be very revealing.

RESPONSE: We thank the reviewer for their positive comments on our work and the interesting suggestions for experiments, which we have responded to below.

A key limitation of the manuscript is that all of the experiments were done with a single B cell line, the chicken DT-40 B cell line. Confirming some of the key findings in primary mouse B cells or a mammalian B cell line would better allow one to generalize the findings to all B cells.

RESPONSE: This point is addressed in the specific response to point 3.

The manuscript would also be strengthened by linking INPP5B to a longer-term readout of B cell activation (e.g. transcription factor activation or BCR-induced apoptosis).

RESPONSE: This point is addressed in the specific response to point 4.

Finally, the experiments employing immobilized anti-Ig are very informative but the physiological relevance would be greatly increased by showing that INPP5B depletion alters B cell spreading and BCR microcluster formation in response to membrane-bound anti-Ig. A situation where the surrogate antigen is mobile would better reflect the process of B cell activation by an antigen-presenting cell. If the authors are in a position to do such experiments, it would be an important extension of the work.

RESPONSE: This point is addressed in the specific response to point 5.

Specific suggestions for further experimentation:

1. Effects of INPP5B depletion on BCR signaling: It is not clear whether INPP5B depletion impairs most BCR signaling responses or only a select subset of downstream responses that are regulated by PI(4,5)P2 and PIP3. This is further complicated by positive and negative feedback loops and by actin-dependent BCR clustering, which amplifies BCR signaling. Hence, a more comprehensive analysis of BCR signaling is warranted.

a) To assess proximal/initial BCR signaling, CD79 phosphorylation (and perhaps also Syk phosphorylation) should be assessed. The commercial pCD79a antibody may work for chicken B cells. Alternatively, if the cells are solubilized with digitonin, mIg-CD79a/b complexes are preserved and could be immunoprecipitated via the stimulating anti-Ig antibody, and then probed with anti-pTyr.

RESPONSE: Unfortunately, we could not analyze CD79A phosphorylation due to lack of antibody cross-reactivity. To assess proximal signaling we therefore looked at phosphorylation of Syk, which binds to the ITAMs on CD79 following phosphorylation by SRC-family kinases (including Lyn and BLK). Interestingly, Syk phosphorylation was unaffected upon INPP5B depletion, suggesting proximal signaling is unaltered. However, as the reviewer alludes to below, feedback loops may complicate the interpretation of this result. We also analyzed CD19 phosphorylation (mediated by Lyn and Syk), and found it was compromised upon INPP5B depletion, suggesting that proximal signaling may indeed be altered. We include the new data on Syk and CD19 phosphorylation in Figure 2E, and discuss the possible interpretation of these results in the revised manuscript text. A more global analysis of signaling across the full timescale of the BCR response would certainly be very interesting, for example using a phosphoproteomics approach, but we feel this lies beyond the scope of the current study.

b) Immunoblotting total cell lysates with anti-pTyr would provide an overall sense of whether BCR signaling responses are globally or selectively reduced in INPP5B-depleted cells. Because, altering phosphoinositide levels could affect feedback loops that amplify or limit BCR signaling, a dose-response for the anti-Ig stimulating antibody (just the 3-minute time point would be sufficient) would reveal whether INPP5B depletion alters the sensitivity of BCR signaling.

RESPONSE: We have performed a dose-response experiment using increasing amounts of anti-Ig at 5 minutes (the shortest time allowing a reliable experiment), and analyzed total pTyr, as suggested by the reviewer. Interestingly, the global pTyr response was not markedly altered by INPP5B depletion at all anti-Ig concentrations. However, one major band was increased in intensity, and several other bands appeared to be less phosphorylated in INPP5B-depleted cells. Hence, although we do see changes, in terms of the overall abundance of pTyr-modified proteins we do not see a significant difference in INPP5B-depleted cells. Further work would be required to identify the pTyr proteins that are altered upon INPP5B depletion. Considering this limitation, we did not include the new data in the manuscript, but include it here for the reviewers (Reviewer Figure 1).

c) Figure 3F shows that fewer cells undergo Ca²⁺ oscillations when INPP5B is depleted and that the oscillations occur at a reduced rate. Is PLC γ 2-mediated PI(4,5)P₂ hydrolysis impaired in the INPP5B-depleted cells? Hence, it would be of interest to measure levels of IP₃ and DAG, especially since INPP5B may also act on IP₃. On page 12, in the discussion, the authors do mention that loss of INPP5B prevents PI(4,5)P₂ hydrolysis in other systems.

RESPONSE: To more directly analyze PLC γ 2 activity, we have measured DAG levels using mass spectrometry. This revealed a reduction of DAG mass in the INPP5B-depleted cells, supporting the assertion that PLC γ 2 activity is reduced upon INPP5B depletion. We also tried to assess IP₃ using a commercially available ELISA kit, but were unable to reliably detect this inositol phosphate in cell extracts. Further experiments would be required to assess IP₃ levels, which would probably necessitate using a different means of analysis e.g.

metabolic labelling and chromatography, which is likely not to be trivial. Based on our observations, we believe that INPP5B is affecting calcium signaling by reducing PLC γ 2 activity downstream from the BCR, although we cannot formally exclude the possibility that it may also act directly on soluble IP3. We have included the new DAG data in Figure 2F.

2. Effects of INPP5B depletion on key actin regulators that are targets of BCR signaling: In addition to cofilin and ezrin, the altered phosphoinositide levels and impaired BCR signaling in INPP5B-depleted cells likely affects multiple nodes in the actin regulatory network. BCR engagement stimulates activation of the Rac2, Cdc42, and Rap1 GTPases, all of which are essential for BCR-induced actin turnover and remodeling. It would be good to assess the activation of these central regulators of actin dynamics using effector pull-down assays, as done for Ras.

RESPONSE: It is certainly possible INPP5B may alter actin dynamics via modulation of a number of actin regulators. As suggested by the reviewer, we have analyzed the levels of active Rap1 and Cdc42 using effector pull-down assays. This showed an increase in active Cdc42 levels upon INPP5B depletion, suggesting that Cdc42 is likely to contribute to the phenotype we observe in the INPP5B-depleted cells. The mechanism underlying this increase in Cdc42 activity is currently unclear, but possible candidates are Vav or DOCK8, Cdc42 GEFs involved in BCR signaling that is activated by plasma membrane PI(4,5)P₂. We did not observe any changes in Rap1 activity upon INPP5B depletion, which suggests Rap1 is not relevant for the changes we see in INPP5B-depleted cells. We were unable to analyze Rac2 in DT40 cells due to poor sensitivity of the antibody. The new Cdc42 and Rap1 data have been included in Figure 6E and Supplementary Figure 4C, respectively.

3. Confirming several of the key findings in primary mouse B cells or mouse/human B cell lines is important. LPS-stimulated primary mouse B cells can be readily transfected with siRNA and spread well on immobilized anti-Ig. Alternatively, a mouse or human B cell line could be transfected with control versus INPP5B siRNA.

RESPONSE: We have analyzed INPP5B in both primary human B cells and in the Ramos human B cell line, which is commonly used to study BCR responses in vitro. We analyzed the BCR response by measuring cell spreading on Ig-coated coverslips, which is dependent on both BCR signaling and downstream actin dynamics, and hence a good readout for both processes. To investigate INPP5B in Ramos cells, we knocked the protein out using CRISPR/Cas9, and could show that cell spreading was markedly reduced in the INPP5B knock-out cells. The knock-out studies were complemented by chemical inhibition of INPP5B in the Ramos cells, using the YU142670 compound, which also inhibited cell spreading. In primary B cells, we could also show that chemical inhibition of INPP5B reduces cell spreading on Ig-coated coverslips. Together, we believe these results support a conserved role for INPP5B in the BCR response in human cells. The human cell data is included in a new figure (Figure 8).

4. It would be good to show that the reduced BCR early signaling in INPP5B-depleted cells have functional consequences and impact longer-term responses that relate to B cell activation or selection.

a) If phospho-specific antibodies that recognize the chicken proteins exist, it should be possible to assess the phosphorylation of NF- κ B, NF-AT, or FOXO1 in control versus INPP5B-

depleted DT-40 cells after BCR engagement. If such reagents do not exist, this could be investigated in mouse or human B cell lines transfected with control versus INPP5B siRNA.

RESPONSE: We have blotted for phospho-FOXO1 and phospho-NFAT. While NFAT phosphorylation was unaffected by INPP5B depletion, the levels of phospho-FOXO1 (at Ser256) were increased. Phosphorylation of this residue by AGC family kinases suppresses FOXO1 activity, which helps prevent apoptosis. Increased phosphorylation is therefore consistent with the decrease in apoptosis we see in INPP5B-depleted cells downstream of BCR stimulation (see response to point 4b). We could not analyze NFkB phosphorylation due to unavailability of a suitable antibody. Together the results indicate that phosphorylation of the FOXO1 transcription factor is altered in INPP5B-depleted cells, consistent with changes in distal signaling under these conditions. The new data is shown in a new figure 7C and D.

b) DT40 cells have been reported to undergo apoptosis after treatment with anti-Ig antibodies and this response is modulated by the level of anti-apoptotic Akt activation (Pogue et al. J Immunol 165:1300, 2000). Would the decreased Akt T308 phosphorylation in INPP5B-depleted DT40 cells make them more susceptible to anti-Ig-induced apoptosis, e.g. at lower concentrations of anti-Ig? Or would impaired BCR signaling and increased BCR endocytosis (leading perhaps to less prolonged BCR signaling) result in less apoptosis? These could be simple but informative experiments.

RESPONSE: To assess the longer-term consequences of INPP5B loss, we have measured apoptosis in the INPP5B-depleted cells, as the reviewer has suggested. We find that depletion of INPP5B results in less BCR-induced apoptosis, which fits with reduced BCR signaling in these cells. Thus, although Akt T308 phosphorylation is decreased, the cells are not more susceptible to apoptosis. We see the converse, suggesting a more widespread impairment of the BCR signaling response. The new data is included in a new figure (Figure 7A and B).

5. If it is possible to do so, some initial experiments showing that INPP5B depletion alters immune synapse formation (B cell spreading, BCR microcluster formation, cSMAC formation) in response to membrane-bound anti-Ig would address the physiological relevance of the findings obtained using immobilized anti-Ig.

RESPONSE: We agree that studying the BCR response using membrane-localized anti-Ig would be very worthwhile. We are currently setting up this technology in the lab, but it is not established and we are not in a position to do these experiments. It is certainly something we would like to do in the future, but feel that it lies beyond the scope of the current study.

Minor comments to address in the text:

1. It would be good to replace the term "BCR activation" in the title (and elsewhere) with "BCR signaling". The term "BCR activation" is more often used to describe the dissociation of autoinhibited BCR dimers (as in Reth's Dissociation-Activation Model), followed by phosphorylation of the BCR ITAMs.

RESPONSE: As suggested we have changed the text from "BCR activation" to "BCR signaling" throughout.

2. Page 7. What is the evidence that capping is important for, or amplifies, BCR signaling? Classically, capping was considered to be a precursor to large-scale BCR endocytosis at later time points (e.g. 30 min).

RESPONSE: The relevant text on page 7 has been changed to refer to clustering more generally as opposed to capping.

3. Page 7. With regard to Fig. 3A, the authors should state that Akt T308 phosphorylation was significantly reduced at 5 min but not at 15 min or 3 min. This raises the possibility that this response is "delayed", more so than "reduced".

RESPONSE: The text describing Fig 2A has been changed to indicate that phosphorylation is delayed.

4. Page 7 and page 10. The term "surface antigen" should be avoided. "Immobilized anti-Ig" or "coverslip-bound anti-Ig" would be more appropriate". Reth and others have shown that anti-Ig Abs that bind to the Fc regions of mIg activate BCRs in a fundamentally different manner than ligands that bind to the antigen-binding site in the mIg subunit of the BCR.

RESPONSE: We thank the reviewer for pointing out this important distinction. We have changed the text accordingly throughout to immobilized anti-IgM or anti-IgM coated coverslip.

5. Is it known whether INPP5B associates constitutively with the inner face of the plasma membrane or whether it is recruited to the plasma membrane by BCR signaling or recruited to nascent BCR clusters? This is a point that could be discussed with regard to the clearance of PI(4,5)P₂ that precedes the appearance of larger BCR microclusters.

RESPONSE: This is a really good point. Currently we do not know the extent of INPP5B recruitment to the B cell plasma membrane before or after stimulation, or the degree of any enrichment in BCR clusters. This is something we are certainly interested in and are actively investigating. Work in other cells suggest INPP5B is cytosolic under steady state (Itzhak et al, eLife, 2016), but the situation in B cells is not known. Our results on cofilin activity, which showed effects of INPP5B depletion prior to BCR stimulation, would suggest a constitutive role for the protein at the plasma membrane. Hence, INPP5B may be recruited to the membrane independently of BCR clustering to modulate cortical actin, which may be important for tonic BCR signaling or other actin-dependent processes. We have added new text to the discussion that describes this possibility. Additional studies will be required to explore this further, and look at the extent of constitutive recruitment versus recruitment to BCR clusters upon stimulation.

6. Page 12. The authors mention that loss of INPP5B prevents PI(4,5)P₂ hydrolysis in other systems. Have these other publications suggested a mechanism by which INPP5B promotes PLC γ -mediated hydrolysis of PI(4,5)P₂?

RESPONSE: INPP5B acts directly on PI(4,5)P₂, and there has not been any study showing influence of INPP5B upon PLC-mediated hydrolysis of this lipid. There is also no known physical association of the enzymes. We therefore believe that the effect of INPP5B depletion upon PLC γ 2 is indirect, as a consequence of reduced BCR clustering and signaling.

7. Page 13. Is it clear that the reduced levels of PIP₃ in INPP5B-depleted cells are due to attenuated PI3K signaling? How would INPP5B, which dephosphorylates PI(4,5)P₂ and

perhaps also PIP3, contribute to activating PI3K or preserving PIP3 levels via other mechanisms?

RESPONSE: We believe that INPP5B is acting indirectly to reduce PIP3 levels, via altered BCR signaling. Support for this interpretation comes from data where we measured PIP3 levels in response to stimulation of B cells with either growth factor (insulin) or cytokine (SDF1). In both cases, depletion of INPP5B had no effect on PIP3 abundance, which strongly suggest that INPP5B is affecting PIP3 levels via the BCR and not because of a more general role in PIP3 signaling e.g. via PI3-kinase or PTEN. We include the new PIP3 data for the reviewers (Reviewer Figure 2).

8. Page 14. Ref. 56 refers to an INPP5B knockout mouse. It might be worthwhile to refer to this in the introduction and add that immune cell development and function has not been studied in these mice (or if it has, to summarize the findings).

RESPONSE: Thank you to the reviewer for this suggestion. The INPP5B mouse has not been studied in terms of the immune system, and we have now included this information in the text of the Discussion.

Reviewer #2 (Comments to the Authors (Required)):

"The inositol 5-phosphatase INPP5B regulates clustering and activation of the B cell receptor" focuses on the mechanisms of antigen-driven F-actin remodeling and BCR clustering. Using an elegant auxin-induced degron system, Droubi et al. find that DT40 B cells lacking INPP5B exhibit defects in synaptic F-actin remodeling, antigen capping, and BCR signaling. They go on to provide evidence that the catalytic activity of INPP5B is critical for its function in this context. Finally, they present imaging experiments and biochemical analyses that a) indicate that INPP5B dependent F-actin clearance is required for BCR cluster formation and b) suggest that INPP5B remodels F-actin via cofilin and ezrin, two established regulators of the synaptic cytoskeleton in B cells. This paper is of interest to the readers of JCB because it reveals a previously unappreciated mechanism of B cell activation and synapse assembly, with potentially important implications for the therapeutic control of B cell function. The data are convincing, and the authors have done a good job of explaining and interpreting their results. I have only a few concerns, which the authors should be able to address in a revision.

RESPONSE: We thank the reviewer for their positive comments on our work, and have responded to their specific points below.

1) The most obvious short-coming of the study is that it relies completely on the DT40 cell line. The authors should confirm key results (such as the INPP5B loss-of-function phenotype) in primary B cells.

RESPONSE: A similar comment was made by reviewer 1. To address this point, we have analyzed INPP5B in both primary human B cells and in the Ramos human B cell line, which is commonly used to study BCR responses *in vitro*. We analyzed the BCR response by measuring cell spreading on Ig-coated coverslips, which is dependent on both BCR signaling and downstream actin dynamics, and hence a good readout for both processes. To investigate INPP5B in Ramos cells, we knocked the protein out using CRISPR/Cas9, and could show that cell spreading was markedly reduced in the INPP5B knockout cells. The knock-out studies were complemented by chemical inhibition of INPP5B in the Ramos cells,

which also inhibited cell spreading. In primary B cells, we could show that chemical inhibition of INPP5B reduces cell spreading on Ig-coated coverslips. Together, we believe these results support a conserved role for INPP5B in the BCR response in human cells. The human cell data is included in a new figure (Figure 8).

2) It is perplexing (but also quite interesting) that INPP5B is absolutely required for PIP2 depletion in the early synapse, but that PLC γ , which is also recruited to the synapse and hydrolyzes PIP2, is not. Perhaps the two enzymes are differentially localized? This is something the authors could address quite easily using fluorescently labelled constructs or immunocytochemistry.

RESPONSE: The functional relationship between PLC γ 2 and INPP5B is an interesting question, and was also commented on by reviewer 1. We were unable to perform localization studies, as suggested, since preliminary work on GFP-INPP5B failed to give a clear-cut localization, at least using an over-expression approach. We also tried endogenously tagging INPP5B but could not visualize the tagged protein, likely due to its low abundance. Further work is required to ascertain how these two enzymes relate to each other in the BCR response. It will certainly be worth investigating, but we feel it lies beyond the scope of the current study.

3) Can the anti-correlation between BCR microclusters and PIP2 (Figure 5) be quantitated directly?

RESPONSE: We have measured this alternatively by quantifying the mean fluorescence intensity (MFI) of PIP2 and BCR foci at the same membrane contact region over time. To draw conclusions about the anti-correlation between the two values, the MFI of PIP2 was divided by the correspondent MFI of BCR quantified at the same membrane contact region. The data were then normalized to the values at "2 min" and expressed as a fold difference over time (which is shown in Figure 5E).

4) The INPP5B G502D phenotype (Figure 3) is more subtle than the full KO phenotype (Figures 1 and 2). This implies at least some phosphatase independent function for INPP5B, correct?

RESPONSE: This is certainly a possibility. INPP5B can bind to various other proteins (Rab GTPases, APPL1, IPIP27, Rac, Cdc42) and so a scaffolding function is certainly possible, which contribute to its role in BCR clustering or signaling. Further studies may shed light on this; for example, by making point mutants deficient in various protein interactions. This is something we are keen to explore in our future work.

5) The low dose LatA strategy shown in Figure S6B is clever. What happens when wild type cells are treated with the same concentrations of LatA?

RESPONSE: Wild-type cells are unaffected at the dose of LatA used in this experiment. We have included this data in a revised version of Figure S5B.

6) In the Discussion, it is stated that INPP5B-induced F-actin remodeling is "mediated through the activation and inactivation of . . . cofilin and . . . ezrin, respectively". The authors do not actually show this, however. They should either qualify this statement (e.g. "our results are consistent with the model in which. . .") or incorporate some sort of ezrin and/or cofilin perturbation approach to test the hypothesis directly.

RESPONSE: The relevant text in the Discussion has been changed to tone down the interpretation, as suggested by the reviewer.

7) It would be interesting to know how the authors think that INPP5B activity is coupled to antigen recognition by the BCR. They should speculate about this in the discussion. Is INPP5B recruited to I α or I β ? Is this recruitment ITAM dependent? Alternatively, might INPP5B lower PIP2 levels constitutively, and could this explain the observed loss-of-function phenotypes?

RESPONSE: This is also an interesting point. We don't know whether and/or how INPP5B is recruited to the BCR upon stimulation. It is certainly possible that INPP5B may have a more constitutive role in PIP2 turnover at the plasma membrane, and indeed the effect we observed on cofilin activation prior to BCR stimulation, points to such a constitutive role. Of course, INPP5B may also be selectively recruited to the BCR upon stimulation, and both scenarios are not mutually exclusive. It is certainly interesting and warrants further study. We have added text to the Discussion to better discuss these possibilities.

Reviewer #3 (Comments to the Authors (Required)):

The activation of B cells is a necessary step in the production of antibodies. B cell responses begin with antigen engagement of the B cell receptor (BCR), which triggers BCR clustering, signalling, immune synapse formation, and antigen endocytosis. All steps of this process require actin remodelling. Despite the importance of actin in regulating these key first events in B cell activation, the mechanisms are not well understood.

This paper identifies a role for the inositol 5 phosphatase INPP5B in each step outlined above. Through a series of well constructed experiments, the authors find that, through hydrolysis of PIP2, INPP5B promotes cofilin-mediated actin severing and ezrin-mediated disruption of connections between the actin cytoskeleton and plasma membrane, which together promote BCR clustering and B cell spreading on antigen-coated substrates.

The paper addresses an important question in B cell biology. It is very clearly written and the findings in my opinion make a very nice contribution to our understanding of B cell activation. Overall I think the conclusions made in the paper are supported by the results, but there are several cases where I think the data analysis methods need to be more clearly communicated. Specific points are listed below.

RESPONSE: We thank the reviewer for their positive comments on our work, and have responded to their specific points below.

Major points / questions

- Control conditions are indicated as 'Vehicle' throughout the text and Figures, but it is not clear to me what the vehicle is. Is it DMSO? In line with this, were control cells treated with auxin at any point to demonstrate that it does not impact cell behaviour in the absence of the auxin degron tag?

RESPONSE: The vehicle is distilled water. We now make this clear in the Experimental Procedures "cell culture". In a series of proof-of-concept experiments, we detected no effect of auxin at the concentration used in WT DT40 cells. Also, OCRL^{Degron/Degron} cells showed no phenotypes (Figure S2), which acted as a control for our INPP5B experiments.

- Figures 1B and 3B show the % cells showing BCR capping. How was this quantified? Was it a manual identification of 'yes' or 'no' for each cell, or was a quantitative approach used?

RESPONSE: BCR capping was quantified using a binary readout method, whereby cells were either generated or failed to generate a BCR cap. A BCR cap was defined by ≥ 2 -fold increase in the total fluorescence intensity of BCR membrane staining at one side of the cell opposite to the other side (Reviewer Figure 3). The BCR capping response was expressed as a percentage of cells scored successful for BCR capping. We have added this to the Experimental Procedures "image analysis".

- It would be helpful to have a bit more information about the quantitation in Figure 5. Figures 5C and 5D show the mean fluorescence intensity of PIP2 and BCR foci, respectively. Are the foci in each figure the same spot, so that the data show the decrease in PIP2 signal and increase in BCR signal over time in the same small region? Also, Figure 5E shows the PIP2/BCR intensity ratio. Does the ratio take into account the mean fluorescence intensity throughout the entire synapse, or is it the ratio at the level of the foci?

RESPONSE: In Figure 5C and 5D, the mean fluorescence intensity (MFI) of PIP2 and BCR foci were quantified at the same membrane contact region over time. In Figure 5E, the MFI of PIP2 was divided by the MFI of BCR quantified at the same membrane contact region.

- Figure 6B shows the analysis of F-actin clearance in B cell synapses. Do the data indicate the percentage of cells in each condition that had cleared actin from the cell centre? If yes, how was this determined? Even in the control cells in Figure 6A, the actin distribution looks relatively homogeneous by 6 minutes, rather than enriched in the periphery and depleted in the middle.

RESPONSE: Sorry for the confusion arisen by our labelling of the X-axis. We now make it clear by adding 'foci'.

Minor points

- Small grammar correction page 13: "... And hence INPP5B may could also impact..." may and could are redundant

- On page 31, the citation Tinevez is not formatted into the bibliography

- Page 35, Figure 5: spelling error, should be INPP5B (not INPPB)

RESPONSE: We have corrected these errors in the revised version of the manuscript.

Reviewer Figure 1: Total pTyr is unaltered in INPP5B-depleted cells.

(A) Protein extracts from INPP5B^{Degron/Degron} cells stimulated with increasing concentration of anti-IgM (for 5min) were blotted for phosphotyrosine. Quantification of two such experiments by densitometry is shown in **(B)**. Total pTyr in each condition was assessed by quantifying the intensity of the entire lane, and data is expressed as a fold increase in pTyr relative to vehicle at time 0. Dotted lines were overlaid with the blot image to ease visual separation of comparable conditions.

A**B**
Reviewer Figure 2: PIP3 is unaltered in INPP5B-depleted cells in response to insulin or CXCL12 (SDF-1) stimulation. Serum-starved (with or without auxin) INPP5B^{Degron/Degron} cells were stimulated with 100nM insulin (**A**) or 50ng/ml CXCL12 (**B**), prior to analysis by MS. PI(3,4,5)P₃ levels are expressed relative to PI (see methods), and data were combined for C36:2, C38:3, and C38:4 species. Data was analysed by two-way ANOVA, and *p* values were calculated using Sidak multiple comparisons test. Error bars represent SD.

Reviewer Figure 3: Representative confocal images of control cells showing successful BCR capping in **(A)** or unsuccessful (failed) BCR capping events in **(B)**. The charts show the relative fluorescent intensities of BCR staining along the trajectories (white lines). The fluorescence intensity was measured on a projection image of the confocal stack. Cells with a polarized enrichment of BCR staining at one pole of the cell with 2-fold higher than the opposite side are considered to have undergone BCR capping successfully.

Reagents used to generate the data in the reviewer figures

Reagent	Supplier	Identifier
Insulin, from bovine pancreas	Sigma-Aldrich	I6634
Recombinant human SDF1 protein	Abcam	ab9798
Anti-Phosphotyrosine, clone 4G10	Merck Millipore	05-321X

June 27, 2022

RE: JCB Manuscript #202112018R

Prof. Martin Lowe
University of Manchester
Faculty of Biology, Medicine and Health
Michael Smith Building
Oxford Road
Manchester M13 9PT
United Kingdom

Dear Prof. Lowe:

Thank you for submitting your revised manuscript entitled "The inositol 5-phosphatase INPP5B regulates clustering and signaling of the B Cell receptor". We would be happy to publish your paper in JCB pending final revisions necessary to meet our formatting guidelines (see details below). In your final revision, please be sure to address reviewer #2's final minor comment.

A. MANUSCRIPT ORGANIZATION AND FORMATTING:

- 1) Text limits: Character count for Articles is < 40,000, not including spaces. Count includes abstract, introduction, results, discussion, and acknowledgments. Count does not include title page, figure legends, materials and methods, references, tables, or supplemental legends.
- 2) Figures limits: Articles may have up to 10 main text figures.
- 3) * Figure formatting: Scale bars must be present on all microscopy images, including inset magnifications. Molecular weight or nucleic acid size markers must be included on all gel electrophoresis.*
- 4) Statistical analysis: Error bars on graphic representations of numerical data must be clearly described in the figure legend. The number of independent data points (n) represented in a graph must be indicated in the legend. Statistical methods should be explained in full in the materials and methods. For figures presenting pooled data the statistical measure should be defined in the figure legends. Please also be sure to indicate the statistical tests used in each of your experiments (either in the figure legend itself or in a separate methods section) as well as the parameters of the test (for example, if you ran a t-test, please indicate if it was one- or two-sided, etc.). Also, if you used parametric tests, please indicate if the data distribution was tested for normality (and if so, how). If not, you must state something to the effect that "Data distribution was assumed to be normal but this was not formally tested."
- 5) Abstract and title: The abstract should be no longer than 160 words and should communicate the significance of the paper for a general audience. The title should be less than 100 characters including spaces. Make the title concise but accessible to a general readership.

* We suggest the following minor edits to your title:

The inositol 5-phosphatase INPP5B regulates B Cell receptor clustering and signaling

6) Materials and methods: Should be comprehensive and not simply reference a previous publication for details on how an experiment was performed. Please provide full descriptions in the text for readers who may not have access to referenced manuscripts.

7)* Please be sure to provide the sequences for all of your primers/oligos and RNAi constructs in the materials and methods. You must also indicate in the methods the source, species, and catalog numbers (where appropriate) for all of your antibodies. * Please also indicate the acquisition and quantification methods for immunoblotting/western blots. *

8) Microscope image acquisition: The following information must be provided about the acquisition and processing of images:
a. Make and model of microscope

- b. Type, magnification, and numerical aperture of the objective lenses
- c. Temperature
- d. Imaging medium
- e. Fluorochromes
- f. Camera make and model
- g. Acquisition software
- h. Any software used for image processing subsequent to data acquisition. Please include details and types of operations involved (e.g., type of deconvolution, 3D reconstitutions, surface or volume rendering, gamma adjustments, etc.).

10) Supplemental materials: There are strict limits on the allowable amount of supplemental data. Articles may have up to 5 supplemental figures. Please also note that tables, like figures, should be provided as individual, editable files. A summary of all supplemental material should appear at the end of the Materials and methods section.

13) ORCID IDs: ORCID IDs are unique identifiers allowing researchers to create a record of their various scholarly contributions in a single place. At resubmission of your final files, please consider providing an ORCID ID for as many contributing authors as possible.

Please note that JCB now requires authors to submit Source Data used to generate figures containing gels and Western blots with all revised manuscripts. This Source Data consists of fully uncropped and unprocessed images for each gel/blot displayed in the main and supplemental figures. Since your paper includes cropped gel and/or blot images, please be sure to provide one Source Data file for each figure that contains gels and/or blots along with your revised manuscript files. File names for Source Data figures should be alphanumeric without any spaces or special characters (i.e., SourceDataF#, where F# refers to the associated main figure number or SourceDataFS# for those associated with Supplementary figures). The lanes of the gels/blots should be labeled as they are in the associated figure, the place where cropping was applied should be marked (with a box), and molecular weight/size standards should be labeled wherever possible. Source Data files will be made available to reviewers during evaluation of revised manuscripts and, if your paper is eventually published in JCB, the files will be directly linked to specific figures in the published article.

B. FINAL FILES:

****The license to publish form must be signed before your manuscript can be sent to production. A link to the electronic license to publish form will be sent to the corresponding author only. Please take a moment to check your funder requirements before choosing the appropriate license.****

Thank you for this interesting contribution, we look forward to publishing your paper in Journal of Cell Biology.

Sincerely,

Ana-María Lennon-Dumenil, PhD
Monitoring Editor

Andrea L. Marat, PhD
Senior Scientific Editor

Journal of Cell Biology

Reviewer #1 (Comments to the Authors (Required)):

The authors should be commended for doing an outstanding job addressing my previous comments as well as those of the other reviewers. The new data that have been added greatly strengthen the manuscript, including providing a more thorough analysis of the effects of INPP5B loss/inhibition on BCR signaling, BCR-induced spreading in another cell line as well as primary human B cells, and a downstream biological response (apoptosis). This is an exceptionally thorough and well done body of work that adds a new player to the BCR signaling diagram and which will be a valuable addition to the literature.

Reviewer #2 (Comments to the Authors (Required)):

The authors have addressed my concerns.

I have only one question. With reference to the new FOXO1 data, the authors state that FOXO1 phosphorylation "promotes cell cycle arrest and/or apoptosis". If this is the case, and they observe more FOXO1 phosphorylation in INPP5B-depleted cells, then why do INPP5B-depleted cells exhibit less apoptosis?

Reviewer #3 (Comments to the Authors (Required)):

I would like to thank the authors for their efforts to address the reviewers' concerns. The new results greatly strengthen the paper, which will be a very nice contribution to the field. I'm happy to recommend the manuscript for publication.